

# Pairing symmetry and fermion projective symmetry groups

**Xu Yang$^\star$, Sayak Biswas$^\dagger$, Shuangyuan Lu$^\dagger$, Mohit Randeria and Yuan-Ming Lu**

Department of Physics, The Ohio State University, Columbus, OH 43210, USA

$\star$ yangxusolidstate@gmail.com

## Abstract

The Ginzburg-Landau (GL) theory is very successful in describing the pairing symmetry, a fundamental characterization of the broken symmetries in a paired superfluid or superconductor. However, GL theory does not describe fermionic excitations such as Bogoliubov quasiparticles or Andreev bound states that are directly related to topological properties of the superconductor. In this work, we show that the symmetries of the fermionic excitations are captured by a Projective Symmetry Group (PSG), which is a group extension of the bosonic symmetry group in the superconducting state. We further establish a correspondence between the pairing symmetry and the fermion PSG. When the normal and superconducting states share the same spin rotational symmetry, there is a simpler correspondence between the pairing symmetry and the fermion PSG, which we enumerate for all 32 crystalline point groups. We also discuss the general framework for computing PSGs when the spin rotational symmetry is spontaneously broken in the superconducting state. This PSG formalism leads to experimental consequences, and as an example, we show how a given pairing symmetry dictates the classification of topological superconductivity.

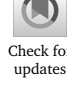
## Contents

$^\dagger$ These authors contributed equally to the development of this work.

# 1 Introduction

One of the most fundamental characterizations of a superconductor or a paired superfluid is the symmetry of its pair wavefunction. The standard way of describing pairing symmetry is in terms of the irreducible representations (irreps) of the *normal state* symmetry group $\mathcal{G}_0$ which constrains the form of the Ginzburg-Landau (GL) free energy functional [1–4]. $\mathcal{G}_0$ can be written as

$$\mathcal{G}_0 = G_0 \times U(1) = \begin{cases} X_0 \times SO(3)_{\text{spin}} \times U(1), & \text{Weak SOC}, \\ X_0 \times U(1), & \text{Strong SOC}, \end{cases} \tag{1}$$

where $X_0$ is the crystalline point group, and SOC denotes spin-orbit coupling. At a second order phase transition, the superconductor spontaneously breaks global charge $U(1)$ symmetry as the system condenses into a particular irrep of the normal state symmetry group. In general, the group of unbroken symmetries in the superconducting phase, $G \subseteq \mathcal{G}_0$. For example, $G = X \times SO(3)_{\text{spin}}$ for a singlet superconductor with weak SOC, where $X \subseteq X_0$ is the point group symmetry preserved in the superconductor. In the presence of a strong SOC we have $G = X$ with $X \subseteq X_0$ being the unbroken point group of the superconductor.

Essentially all of the phonon-mediated superconductors (SCs) exhibit singlet "$s$-wave" pairing, where the superconducting (SC) state transforms according to the trivial representation of $X_0$. But superfluid $^3$He [5] and many quantum materials, including the heavy fermion SCs [6], the high $T_c$ cuprates [7], and $Sr_2RuO_4$ [8], condense into nontrivial irreps.

In this paper, we wish to focus on the relation between pairing symmetry and the symmetry of the Hamiltonian describing the *fermionic excitations in the superconducting state*. At the mean field level, one focuses on the Bogoliubov-de Gennes (BdG) Hamiltonian, but the fermionic symmetry analysis applies equally beyond the BdG framework where one needs to take into account interactions between quasiparticles. The approach we develop here will allow us to gain new insights that go beyond the (bosonic) GL theory.

Examples of questions which this formalism would shed light on include: (a) the relation between pairing symmetry and topology, as the K-theory classification [9–11] of non-interacting topological SCs is based on the BdG Hamiltonians, (b) how interactions between quasiparticles for various pairing symmetries impact the classification of interacting topological SC phases [11–14], (c) the relation between pairing symmetry and excitations in topological defects such as Majorana zero modes trapped in vortices [15–19], and (d) whether

new probes of electronic excitations can provide insight into the pairing symmetry [20]. We discussed question (a) in section 4.4 of the manuscript. We will return to other questions in subsequent papers.

Here, we first show how starting with the pairing symmetry, together with the crystalline symmetries that dictate the normal state electronic structure, we can derive the projective symmetry group (PSG) [21] for the fermionic excitations in the SC state. We first focus on the cases where the superconductor shares the same spin rotational symmetry as the normal state. We present an exhaustive classification of the SC state PSG corresponding to every allowable pairing symmetry for the 32 crystalline point groups with and without SOC. When confronted with a new superconductor, we would like to use these results in the "reverse" direction, namely, how can we deduce the possible pairing symmetry, given fermionic properties in the SC state? Mathematically, the map from the pairing symmetry to the SC state PSG is, in general, neither injective nor surjective, and thus it cannot be inverted. Nevertheless, we show below that the SC state PSG does constrain to a considerable extent the possible pairing symmetries. We also present numerous examples that serve to illustrate our general results.

To describe the symmetries of the fermionic Hamiltonian we need (i) to focus on the *superconducting state* symmetry group $G$ as distinct from the *normal state* $G_0$ relevant for GL theory, and (ii) to take into account fermion parity $(-1)^{\hat{F}}$, where $\hat{F}$ is the total number of fermions in the system. Let us discuss each of these points in turn.

On general grounds, the SC state symmetry group $G$ is a subgroup of the normal state $G_0$. If the irrep into which the GL theory condenses is one-dimensional, then in fact $G = G_0$. While this is obvious for the trivial $A_1$ representation, an example may be useful to illustrate why this is true quite generally. Consider the $d_{x^2-y^2}$ pairing state in the cuprates that transforms according to the $B_{1g}$ irrep of the tetragonal symmetry group $D_{4h}$. The pair wavefunction changes sign under a $\pi/2$ rotation, and one might naively think that this breaks $C_4$ down to $C_2$. However, one can compensate for this minus sign by having the fermion operators pick up an $e^{i\pi/2}$ phase under $C_4$ and thus have the electronic Hamiltonian retain the full symmetry of the normal state. We will see a generalization of this at play in the analysis later in section 2.

On the other hand, if the irrep has a dimension $> 1$, then one needs to solve the GL equations to find the SC state that minimizes the free energy. Then the SC state state symmetry is lower than that in the the normal state, and $G$ is a proper subgroup of $G_0$. For example, ${}^3$He is a $p$-wave, triplet superfluid, corresponding to the $L = 1, S = 1$ irrep of the normal state symmetry group $G_0 = SO(3)_{\text{orbital}} \times SO(3)_{\text{spin}}$. Depending on external parameters various superfluid states are stabilized, and in the $B$-phase of ${}^3$He, for instance, $G_0$ is broken down to $G = SO(3)_{L+S}$ [22]. We will discuss a general framework to understand the PSG of fermion excitations in any superconductor in Section 5, where the superconductor can spontaneously break the normal-state spin rotational symmetry.

The second point above related to fermion parity may seem trivial: it enforces that a Hamiltonian can only have terms with an even number of fermion operators. It leads, however, to the important mathematical structure of a projective symmetry group (PSG) $G_f$ acting on the many-body Hilbert space. In Section 2, we discuss in detail how $G_f$ is built as a central extension of $G$ by the fermion parity group $\mathbb{Z}_2^F$.

The rest of the paper is organized as follows. In Section 3 we show how the fermion PSG $G_f$ can constrain the pairing symmetry of the SC state, applying the framework to all 32 point groups (see Table 7) and demonstrating it by a few examples in section 4. We further discuss how the PSG determines topological properties of the SC in section 4.4. While sections 2-3 focus on the cases where the normal state and the SC state shares the same spin rotational symmetries, in section 5.1 we describe a generic theory framework that applies to all superconductors, and further demonstrate its power in the examples of A- and B-phases of superfluid ${}^3$He in section 5.2. Finally we conclude in section 6 with a discussion on how the fermion PSG

in SCs discussed here differs from the PSG first introduced in quantum spin liquids [21, 23], and an outlook to future studies.

# 2 Characterization of broken symmetries in a superconductor

## 2.1 Projective symmetry group and projective representation

Any Hamiltonian must conserve fermion parity $(-1)^{\hat{F}}$ even if it does not conserve particle number $\hat{F}$, as, for instance, in the presence of pairing. The fermion symmetry group $G_f$ acting on the many-body Hilbert space of fermions is a projective symmetry group (PSG). Mathematically, $G_f$ is a central extension of the bosonic symmetry group $G$ in the SC state by the fermion parity group $\mathbb{Z}_2^F = \left\{ (\pm 1)^{\hat{F}} \right\}$. This may be written as a short exact sequence

$$1 \to \mathbb{Z}_2^F \to G_f \to G \to 1, \tag{2}$$

where $\mathbb{Z}_2^F$ is in the center of $G_f$. Thus fermion parity commutes with all elements of $G_f$ and the quotient group $G_f / \mathbb{Z}_2^F = G$.

Let us denote by $\hat{g}$ the operator corresponding to the group element $g \in G$ that acts on Hilbert space. In general it could be unitary or anti-unitary. The group $G_f$ is then the set $\left\{ (\pm 1)^{\hat{F}} \hat{g} \mid g \in G \right\}$ with the product rule between $(\eta_1)^{\hat{F}} \hat{g}$ and $(\eta_2)^{\hat{F}} \hat{h}$ (with $\eta_i = \pm 1$) given by

$$\left[ (\eta_1)^{\hat{F}} \hat{g} \right] \left[ (\eta_2)^{\hat{F}} \hat{h} \right] = [\eta_1 \, \eta_2 \, \omega(g,h)]^{\hat{F}} \, \widehat{g\,h}, \tag{3}$$

$\omega$ called the 2-cocycle is a function $\omega : G \times G \to \{+1, -1\}$ that satisfies[1]

$$\omega(g,h)\,\omega(gh,k) = \omega(g,hk) \, {}^g\omega(h,k), \tag{4}$$

so that the multiplication is associative, and $\omega(e_G, e_G) = 1$, so that the identity element is well defined. Each inequivalent cocycle furnishes a distinct projective symmetry group. Thus PSGs are characterized by classes of inequivalent cocycles $[\omega]$ which form the second cohomology group $\mathcal{H}^2(G, \mathbb{Z}_2)$.

As an example, consider time reversal symmetry where $G = \mathbb{Z}_2^T = \{\mathbb{1}, T\}$. In this case, $\mathcal{H}^2(\mathbb{Z}_2, \mathbb{Z}_2) = \mathbb{Z}_2$ and there are two PSGs characterized by the two inequivalent cocycles: (1) $\omega(T, T) = 1$ in which case $\hat{T}^2 = 1$, and (2) $\omega(T, T) = -1$ where $\hat{T}^2 = (-1)^{\hat{F}}$. In the first case $G_f = \mathbb{Z}_2 \times \mathbb{Z}_2$ while in the second $G_f = \mathbb{Z}_4$. Physically, the action of the different PSGs on the even particle number subspace is the same as that of the bosonic group $G$. The distinction appears in how $G_f$ acts on the odd particle number subspace, in particular, the single particle subspace.

In general, one could have both unitary and anti-unitary symmetries but in this paper we will focus on *unitary* operators $\hat{g} \in G_f$, under which the fermion annihilation operator transforms as

$$\hat{g} \, \hat{c}_{\mathbf{k}\,\alpha} \, \hat{g}^{-1} = [U^g(\mathbf{k})]^\dagger_{\alpha\beta} \, \hat{c}_{g\mathbf{k}\,\beta}, \tag{5}$$

where $\mathbf{k}$ is the (crystal) momentum, and the $\alpha$ labels spin, orbital/sublattice/band degrees of freedom (d.o.f.). Using $(-1)^{\hat{F}} \hat{c}_{\mathbf{k}\,\alpha} (-1)^{\hat{F}} = -\hat{c}_{\mathbf{k}\,\alpha}$ and eq. (3), we find that

$$U^g(h\,\mathbf{k}) \, U^h(\mathbf{k}) = \omega(g,h) \, U^{gh}(\mathbf{k}). \tag{6}$$

The $U^g$'s thus form a projective representation of $G$ with coefficients in $\{\pm 1\}$. Equivalently, one can regard $\{\pm U^g \mid g \in G\}$ as a linear representation of $G_f$ with $(-1)^{\hat{F}}$ represented by $-\mathbb{1}$.

---

[1]Here we define ${}^g\omega(h,k) = \hat{g}\,\omega(h,k)\hat{g}^{-1}$, which is $\omega(h,k)$ if $g$ is a unitary or $\omega^*(h,k)$ if $g$ is an anti-unitary symmetry.

## 2.2 Pairing symmetry and projective representations

To be concrete, let us focus on the BdG Hamiltonian

$$\hat{H} = \hat{H}_0 + (\hat{H}_{\text{pair}} + \text{h.c.}), \tag{7}$$

where

$$\hat{H}_0 = \sum_{\alpha\beta;\mathbf{k}} \hat{c}_{\mathbf{k}\alpha}^\dagger h_{\alpha\beta}(\mathbf{k})\hat{c}_{\mathbf{k}\beta}, \tag{8}$$

is the kinetic energy that describes the normal state electronic dispersion, and

$$\hat{H}_{\text{pair}} = \sum_{\alpha\beta;\mathbf{k}} \hat{c}_{\mathbf{k}\alpha}^\dagger \Delta_{\alpha\beta}(\mathbf{k})\hat{c}_{-\mathbf{k}\beta}^\dagger, \tag{9}$$

describes the pairing. Fermi statistics dictates that $\Delta_{\alpha\beta}(\mathbf{k}) = -\Delta_{\beta\alpha}(-\mathbf{k})$.

Initially, we restrict ourselves for simplicity to situations where $SO(3)_{\text{spin}}$ is *not* broken spontaneously in the SC state. In this case, the SC state symmetry group $G$ is of the form

$$G = \begin{cases} X \times SO(3)_{\text{spin}}, & \text{Weak SOC}, \\ X, & \text{Strong SOC}, \end{cases} \tag{10}$$

where $X$ is the point group of crystalline symmetries. In either case the pairing order parameter $\Delta(\mathbf{k})$ forms a 1d linear representation of crystalline point group $X$. Moreover the relevant fermionic PSGs are of the form $G_f \simeq (X_f \times SU(2))/\mathbb{Z}_2$ and $G_f \simeq X_f$ for the weak and strong SOC cases respectively where $X_f$ is itself a central extension of $X$ with respect to fermion parity. In the first case, we get an $SU(2)$ as a $\mathbb{Z}_2$ central extension of $SO(3)_{\text{spin}}$ and a quotient by $\mathbb{Z}_2$ is required to take into account the "double- counting" of $\mathbb{Z}_2^F$. It is thus sufficient to look at the central extensions of $X$. Later, in Section 5, we shall present a more general treatment and discuss the case of $^3$He where spin rotation is spontaneously broken in the SC state. In such cases, the fermion symmetry group might have a more complicated form and it is no longer sufficient to look at central extensions of the spatial part alone.

We now discuss *three* different projective representations of $X$ and explore how these are related. First, we begin with $X_f^0 = \left\{ (\pm 1)^{\hat{F}} \hat{g}_0 \mid g \in X \right\}$ the PSG of $X$ that preserves the kinetic part of the BdG hamiltonian i.e., $\hat{g}_0 \hat{H}_0 \hat{g}_0^{-1} = \hat{H}_0$. The fermion operators then transform according to the corresponding projective representation $U_0^g(\mathbf{k})$, defined by

$$\hat{g}_0 c_{\mathbf{k}\alpha} \hat{g}_0^{-1} = [U_0^g(\mathbf{k})]_{\alpha\beta}^\dagger \hat{c}_{g\mathbf{k}\beta}, \tag{11}$$

which preserves the normal state band structure

$$U_0^g(\mathbf{k}) h(\mathbf{k}) [U_0^g(\mathbf{k})]^\dagger = h(g\mathbf{k}). \tag{12}$$

We shall call $X_f^0$ the *normal state PSG* and denote the corresponding 2-cocycle by $\omega_0$. For systems with weak SOC, crystalline symmetries do not act on the spin degrees of freedom and the PSG is trivial in this case $\omega_0(g,h) = 1$ for any elements $g, h \in X$. In the presence of strong SOC the projective representation is non-trivial with operations like two fold rotations and mirror reflections now squaring to fermion parity, $\omega_0(C_2, C_2) = \omega_0(M, M) = -1$. This becomes evident by looking at the forms of the projective representations in the two cases.

$$U_0^g(\mathbf{k}) = \begin{cases} u_{\text{orbital}}^g(\mathbf{k}) \otimes \mathbb{1}_{\text{spin}}, & \text{weak SOC}, \\ u_{\text{orbital}}^g(\mathbf{k}) \otimes e^{i\frac{\theta_g}{2}\hat{n}_g \cdot \vec{\sigma}}, & \text{strong SOC}, \end{cases} \tag{13}$$

where $\hat{n}_g$ and $\theta_g$ are the rotation axis and angle associated with crystalline symmetry operation $g \in X$.

Next, we note that the normal state PSG preserves the pairing term only up to a phase, namely

$$\hat{g}_0 \, \hat{H}_{\text{pair}} \, \hat{g}_0^{-1} = e^{i\Phi_g} \hat{H}_{\text{pair}} \,. \tag{14}$$

The phases $\left\{ e^{i\Phi_g} \mid g \in X \right\}$ form a 1D *linear* representation of $X$, which we call the *pairing symmetry* $\mathcal{R}_{\text{pair}}$. The phases $\Phi_g$'s satisfy the equation

$$\Phi_g + \Phi_h = \Phi_{gh} + 2n\pi \quad (n \in \mathbb{Z}) \,. \tag{15}$$

The pairing matrix $\Delta(\mathbf{k})$ satisfies

$$U_0^g(\mathbf{k})\Delta(\mathbf{k})\left[ U_0^g(-\mathbf{k}) \right]^T = e^{i\Phi_g} \, \Delta(g\mathbf{k}) \,. \tag{16}$$

We see from eq. (14) that the PSG $X_f^0$ that leaves $\hat{H}_0$ invariant, fails to preserve the pairing term. However the situation can be fixed as follows. We modify the transformation of the fermions $\hat{g}' \, c_{\mathbf{k}\alpha} \hat{g}'^{-1} = [\tilde{U}(\mathbf{k})]_{\alpha\beta}^\dagger \, \hat{c}_{\mathbf{k}\beta}$ with

$$\tilde{U}^g(\mathbf{k}) = e^{-i\Phi_g/2} U_0^g(\mathbf{k}) \,. \tag{17}$$

The kinetic part $\hat{H}_0$, which is invariant under $U(1)$ phase rotations, is preserved by the modified transformations as can be seen from (12). The new transformations are also symmetries of the pairing term $\hat{H}_{\text{pair}}$ as $\tilde{U}_g(\mathbf{k})$'s lead to eq. (16) without the phase factor $e^{i\Phi_g}$ appearing on the right-hand side.

We thus define *SC state PSG $\tilde{X}_f$* that preserves the full BdG Hamiltonian by

$$\tilde{X}_f = \left\{ (\pm 1)^{\hat{F}} \, \hat{g}' = (\pm 1)^{\hat{F}} \, e^{-i(\Phi_g/2)\hat{F}} \, \hat{g} \mid g \in X \right\} \,. \tag{18}$$

This PSG is characterized by the 2-cocycle $\tilde{\omega}$.

The last step here is to look at the relation between the normal and the superconducting state PSGs, or equivalently, between their cocycles $\omega_0$ and $\tilde{\omega}$. The phases $\left\{ e^{-i\Phi_g/2} \mid g \in X \right\}$ form a 1D *projective* representation of $X$, which we call $\mathcal{R}_\Phi$. This follows from (15) by observing that $e^{-i\Phi_g/2}e^{-i\Phi_h/2} = (-1)^n \, e^{-i\Phi_{gh}/2}$. From eqn.(17) one concludes that the cocycle $\omega_\Phi$ associated with $\mathcal{R}_\Phi$ satisfies

$$\tilde{\omega}(g,h) = \omega_\Phi(g,h) \, \omega_0(g,h) \,. \tag{19}$$

To summarize, we encountered the following projective representations and their associated cocycles which define the corresponding PSG's:

$$\text{Normal state:} \quad U_0^g(h\,\mathbf{k}) \, U_0^h(\mathbf{k}) = \omega_0(g,h) \, U_0^{gh}(\mathbf{k}) \,, \tag{20a}$$

$$\mathcal{R}_\Phi : \quad e^{-i\Phi_g/2}e^{-i\Phi_h/2} = \omega_\Phi(g,h) e^{-i\Phi_{gh}/2} \,, \tag{20b}$$

$$\text{SC state:} \quad \tilde{U}^g(h\,\mathbf{k}) \, \tilde{U}^h(\mathbf{k}) = \tilde{\omega}(g,h) \, \tilde{U}^{gh}(\mathbf{k}) \,. \tag{20c}$$

Eq. (17) relates the three projective representations and eq. (19) relates their cocycles.

Given the normal state PSG and the pairing symmetry of the SC state, one can use the formalism described above to determine the SC state PSG. This is achieved in the following steps. Pairing symmetry being a 1D linear representation, $\mathcal{R}_{\text{pair}}$ can be read off from the character table of $X$. Taking the square roots of the characters one obtains the 1D projective representation $\mathcal{R}_\Phi$ and its cocycle $\omega_\Phi$. With the normal state PSG known eq. (19) gives the SC state PSG while eq. (17) gives the SC state projective representation explicitly. Thus knowing the pairing symmetry enables us to find the SC state PSG that preserves the BdG Hamiltonian. In the next Section we turn to the inverse problem of constraining the pairing symmetry, knowing the SC state PSG.

# 3 Constraints on the pairing symmetry by the PSG

One longstanding experimental challenge in the field of superconductivity is how to unambiguously determine the pairing symmetry of a superconductor material, based on experimental measurements. Since all fermionic excitations in the superconductor form a linear representation of the SC state PSG $\tilde{X}_f$, the low-temperature physical properties of the superconductor completely depend on the PSG. For example, as will be discussed in section 4.4, the topological properties of the SC phase are determined by the PSG. As a result, it seems plausible to detect the SC state PSG $\tilde{X}_f$ using various experimental probes, which we will clarify in future publications. This observation motivates us to answer the following question: given a SC state PSG $\tilde{X}_f$, what are the pairing symmetries compatible with $\tilde{X}_f$? In other words, how does a given PSG constrain the possible pairing symmetry in a superconductor? The answer to this question will allow us to constrain or even determine the pairing symmetry of a SC, by experimentally detecting its PSG.

Based on the discussions in section 2.2, we can readily derive the constraints on the pairing symmetry by the PSG from relations (17) and (19). Specifically, given a SC state PSG $\tilde{X}_f$ and its associated 2-cocycle $\tilde{\omega}$, we can follow the steps listed below to obtain the possible pairing symmetries $\mathcal{R}_{pair}$ in (14)-(16):

(1) Given the crystalline point group $X$, determine the normal state PSG $X_f^0$ and associated 2-cocycle $\{\omega_0\}$ of the normal-state symmetry transformations $\{U_0^g | g \in X\}$. This only depends on the strength of SOCs in the system.

(2) Compute the 2-cocycle $\{\omega_\Phi\}$ from $\{\omega_0\}$ and $\{\tilde{\omega}\}$ from relation (19).

(3) Obtain all one-dimensional (1d) projective representations $\{\mathcal{R}_\Phi(g) | g \in X\}$ of the crystalline symmetry group $X$ compatible with 2-cocycle $\{\omega_\Phi\}$ obtained in step (2), satisfying

$$\mathcal{R}_\Phi(g)\mathcal{R}_\Phi(h) = \omega_\Phi(g,h)\mathcal{R}_\Phi(gh). \tag{21}$$

(5) For each 1d projective representation $\mathcal{R}_\Phi(g)$ obtained in step (3), compute the 1d linear representation,

$$\mathcal{R}_{pair}(g) = \left[\mathcal{R}_\Phi(g)\right]^{-2}, \tag{22}$$

of the pairing order parameter. The collection of all results $\{\mathcal{R}_{pair}\}$ correspond to all the possible pairing symmetries compatible with the PSG $\tilde{X}_f$.

We have applied our general computational scheme to the case of 32 crystalline point groups for both strong SOCs and neglible (weak) SOCs. Our analysis focuses on SC order parameters that are spatially uniform, where the Cooper pairs condense in a state with zero center of mass momentum. Lattice translations then act trivially on the SC state and leave the BdG Hamiltonian invariant, and it is sufficient for us to focus on point group symmetry alone. Most experimentally relevant systems exhibit spatially uniform pairing (in the absence of strong disorder). It is only in exceptional circumstances – under very limited range of external parameters in a few systems – that that one expects the SC order parameter to spontaneously break translational symmetry, e.g., in FFLO or pair density wave state. In such cases, we would need to investigate space group symmetries which we leave for future investigation. Group cohomology and projective representation calculations are performed using the GAP computer algebra program [24]. The correspondence between fermion PSGs $G_f$ and the representations $\mathcal{R}_{pair}$ of the superconducting order parameter is established for all 32 point groups, and the results are summarized in Table. 7 in Appendix B.

# 4 Examples

We now demonstrate the above formalism for different point groups. In section 4.1 we consider systems with tetragonal symmetry. Cuprates and ruthenates which belong to this category have point group $D_{4h}$. But for instance in cuprates, only the Cu-O plane is relevant for superconductivity and it suffices to consider the point group $C_{4v}$ for the purpose of illustration. In section 4.2 we treat systems with hexagonal symmetry. A discussion of superconductivity on a honeycomb lattice is followed by a remark on how our formalism can be applied to the case of magic angle twisted bilayer graphene. In section 4.3 we discuss superconductivity in transition metal dichalcogenides with trigonal point group $C_{3v}$.

The purpose of these examples is two-fold. First, we present a detailed account of how the table in appendix B is constructed and what information can be extracted from it. Second, we make a direct connection with real physical systems by producing examples of order parameters $\Delta_{\alpha\beta}(\mathbf{k})$ for each 1D irrep (pairing symmetry) of the relevant point group.

As mentioned earlier we shall restrict ourselves to cases where there is no additional breaking of spin rotation symmetry when going from the normal to the SC phase. Examples which do not fit in this category, like superfluid He³, are discussed in the section 5.2.

## 4.1 Tetragonal symmetry

To be concrete, consider a two dimensional square lattice in the $xy$ plane. The relevant crystalline point group is $X = C_{4v}$. The group is generated by a rotation by $\pi/2$ about the $z$-axis, $C_4$ and reflection about a vertical mirror in the $yz$ plane, $\sigma_v$. The action of these operations can be summarized as

$$(x, y, z) \xrightarrow{C_4} (-y, x, z), \tag{23a}$$

$$(x, y, z) \xrightarrow{\sigma_v} (-x, y, z). \tag{23b}$$

The group law is captured by the relations $C_4^4 = e$, $\sigma_v^2 = e$ and $C_4^3 \sigma_v = \sigma_v C_4$. Equivalently the group is generated by the vertical mirror $\sigma_v$ and the diagonal mirror $\sigma_d = \sigma_v C_4$. Since $\sigma_v^2 = \sigma_d^2 = e$, these could have either $+1$ or $-1$ characters in a 1D irrep. Consequently there are four 1D irreps for this group, each labeled uniquely by a tuple of $\sigma_v$ and $\sigma_d$ characters, $\left( e^{i\Phi_{\sigma_v}}, e^{i\Phi_{\sigma_d}} \right)$ taking values $(\pm 1, \pm 1)$. The characters for the other group elements can then be obtained using the group laws. In particular, it follows from $C_2 = (\sigma_d \sigma_v)^2$ that the character for the two-fold rotation in the four 1D irreps is $+1$.

Let us now turn our attention to the possible fermion PSGs for this group. From the group cohomology calculation we have $\mathcal{H}^{(2)}(C_{4v}, \mathbb{Z}_2) = \mathbb{Z}_2^3$, corresponding to eight inequivalent classes of 2-cocycles for this group characterized by the 3-tuple

$$(\omega(C_2, C_2), \omega(\sigma_v, \sigma_v), \omega(\sigma_d, \sigma_d)) = (\pm 1, \pm 1, \pm 1). \tag{24}$$

The eight PSGs are thus distinguished on the basis of whether the two fold rotation, $C_2$ and the two mirrors $\sigma_v$ and $\sigma_d$ square to $\pm 1$.

We are now in a position to explore the connection between the pairing symmetries and fermion PSGs for this group. First consider the case when because of weak spin-orbit coupling there is spin rotation invariance in the normal state. The symmetry operations that preserve the kinetic energy act only on the momentum label, keeping the spin label unaltered. Denoted by superscript 0 these are

$$\hat{C}_2^0 \, \hat{c}_{\mathbf{k}\alpha} \, (\hat{C}_2^0)^{-1} = \hat{c}_{C_2 \mathbf{k}\alpha}, \tag{25a}$$

$$\hat{\sigma}_v^0 \, \hat{c}_{\mathbf{k}\alpha} \, (\hat{\sigma}_v^0)^{-1} = \hat{c}_{\sigma_v \mathbf{k}\alpha}, \tag{25b}$$

$$\hat{\sigma}_d^0 \, \hat{c}_{\mathbf{k}\alpha} \, (\hat{\sigma}_d^0)^{-1} = \hat{c}_{\sigma_d \mathbf{k}\alpha}. \tag{25c}$$

Consequently, the normal state PSG is trivial and

$$(\omega_0(C_2, C_2), \omega_0(\sigma_v, \sigma_v), \omega_0(\sigma_d, \sigma_d)) = (+1, +1, +1). \tag{26}$$

Given the assumption that pairing does not break spin rotation invariance in the superconducting phase, condensation takes place in the singlet channel. This enforces the pair wavefunction to be of the form

$$\Delta_{\alpha\beta}(\mathbf{k}) = \Psi(\mathbf{k})(i\sigma_y)_{\alpha\beta}, \tag{27}$$

where $\alpha$, $\beta$ are spin labels and Pauli exclusion constrains the orbital part of the pair wavefunction to obey $\Psi(-\mathbf{k}) = \Psi(\mathbf{k})$. As has been discussed in detail in previous sections, the phases $\{e^{i\Phi_g}\}$ acquired by the pairing term in (9), when acted upon by the operations in (25), constitute a 1D linear irrep of $C_{4v}$ which we refer to as pairing symmetry $\mathcal{R}_{\text{pair}}$. We also learnt that (25) must be modified by compensating phase rotations so as to make the new transformations symmetries of the BdG hamiltonian.

Different pairing symmetries modify the normal state transformations in (25) differently. When the pairing symmetry is $A_1$, which is the case when say $\Psi(\mathbf{k})$ is a constant $\Psi_0$ independent of $\mathbf{k}$, the normal state transformations already preserve the pairing term and no modification is necessary. The normal and the SC state PSGs are the same in this case. If however $\Psi(\mathbf{k}) = \Psi_0(k_x^2 - k_y^2)$, $\sigma_v$ keeps the pairing term unchanged whereas under $\sigma_d$ (or equivalently under $C_4$) it acquires a negative sign. The pairing symmetry in this case is $B_1$, labeled by $(e^{i\Phi_{\sigma_v}}, e^{i\Phi_{\sigma_d}}) = (+1, -1)$. Eqn. (25c) now has to be modified by a factor of $i$ appearing on the right hand side, i.e, the modified $\sigma_d$ must take $\hat{c}_{\mathbf{k}\alpha}$ to $i\hat{c}_{\sigma_d\mathbf{k}\alpha}$.

For a generic irrep, when the orbital part transforms as

$$\Psi(\mathbf{k}) = e^{i\Phi_g}\Psi(g\mathbf{k}), \tag{28}$$

the compensating phases are the square roots of the characters of the relevant irrep. Denoted with primes, the transformations that preserve the BdG hamiltonian are then

$$\hat{C}_2' \hat{c}_{\mathbf{k}\alpha} (\hat{C}_2')^{-1} = e^{i\Phi_{C_2}/2} \hat{c}_{C_2\mathbf{k}\alpha}, \tag{29a}$$

$$\hat{\sigma}_v' \hat{c}_{\mathbf{k}\alpha} (\hat{\sigma}_v')^{-1} = e^{i\Phi_{\sigma_v}/2} \hat{c}_{\sigma_v\mathbf{k}\alpha}, \tag{29b}$$

$$\hat{\sigma}_d' \hat{c}_{\mathbf{k}\alpha} (\hat{\sigma}_d')^{-1} = e^{i\Phi_{\sigma_d}/2} \hat{c}_{\sigma_d\mathbf{k}\alpha}. \tag{29c}$$

For instance, for $A_1$ and $B_1$ pairing symmetries, $\left(e^{i\Phi_{C_2}/2}, e^{i\Phi_{\sigma_v}/2}, e^{i\Phi_{\sigma_d}/2}\right)$ can be chosen to be $(1, 1, 1)$ and $(1, 1, i)$ respectively.

The resulting SC state PSGs are different across pairing symmetries. For the $A_1$ irrep, the SC state PSG is trivial. With the diagonal mirror now squaring to fermion parity, the SC state PSG for $B_1$ becomes

$$(\tilde{\omega}(C_2, C_2), \tilde{\omega}(\sigma_v, \sigma_v), \tilde{\omega}(\sigma_d, \sigma_d)) = (+1, +1, -1). \tag{30}$$

As elaborated in previous sections, the reason for this is best understood once we recognize that the compensating phases, $\left\{e^{-i\Phi_g/2} \mid g \in X\right\}$ form a 1D projective representation, $\mathcal{R}_\Phi$ of $X$. The corresponding cocycle given by $\omega_\Phi$ could be different for the different pairing symmetries. For example,

$$(\omega_\Phi(C_2, C_2), \omega_\Phi(\sigma_v, \sigma_v), \omega_\Phi(\sigma_d, \sigma_d)) = (1^2, 1^2, 1^2), \quad \text{and} \quad (1^2, 1^2, i^2 = -1), \tag{31}$$

for the $A_1$ and $B_1$ irreps respectively. Pairing symmetry thus dictates $\omega_\Phi$ which through (19) in turn decides the SC state PSG. Table 1 summarizes the results of the above analysis for $C_{4v}$ with weak SOC. For each irrep, we give an example of $\Psi(\mathbf{k})$, show the 1D projective representation of the compensating phases $\mathcal{R}_\Phi$, the cocycle $\omega_\Phi$ and finally the SC state PSG $\tilde{\omega}$.

Table 1: Tetragonal Symmetry ($X = C_{4v}$) with weak SOC. Here $\mathcal{R}_{\text{pair}} \equiv (e^{i\Phi_{\sigma_v}}, e^{i\Phi_{\sigma_d}})$ and $\mathcal{R}_\Phi \equiv (e^{-i\Phi_{C_2}/2}, e^{-i\Phi_{\sigma_v}/2}, e^{-i\Phi_{\sigma_d}/2})$.

| $\mathcal{R}_{\text{pair}}$ | $\Psi(\mathbf{k})$ | $\mathcal{R}_\Phi$ | $\omega_\Phi$ | $\tilde{\omega}$ |
|---|---|---|---|---|
| $A_1 : (+1, +1)$ | $1$ | $(\pm 1, \pm 1, \pm 1)$ | $(+1, +1, +1)$ | $(+1, +1, +1)$ |
| $A_2 : (-1, -1)$ | $k_x k_y (k_x^2 - k_y^2)$ | $(\pm 1, \pm i, \pm i)$ | $(+1, -1, -1)$ | $(+1, -1, -1)$ |
| $B_1 : (+1, -1)$ | $k_x^2 - k_y^2$ | $(\pm 1, \pm 1, \pm i)$ | $(+1, +1, -1)$ | $(+1, +1, -1)$ |
| $B_2 : (-1, +1)$ | $k_x k_y$ | $(\pm 1, \pm i, \pm 1)$ | $(+1, -1, +1)$ | $(+1, -1, +1)$ |

In the presence of strong spin orbit coupling, the transformations that preserve the kinetic energy are combined spatial and spin rotation. A rotation by angle $\theta$ about $\hat{\mathbf{n}}$ transforms the spinor by $e^{-i\frac{\theta}{2}(\hat{\mathbf{n}}\cdot\sigma)}$ while inversion leaves it unaffected. A mirror could be viewed as a combination of inversion and a two fold rotation about an axis perpendicular to the mirror plane. For instance, reflection about the $yz$ mirror plane is then effectively a two-fold rotation about the $x$ axis and would be implemented by $-i\sigma_x$ in the spinor basis. The transformations that preserve kinetic energy are

$$\hat{C}_2^0 \, \hat{c}_{\mathbf{k}\alpha} \, (\hat{C}_2^0)^{-1} = [-i\sigma_z]_{\alpha\beta} \, \hat{c}_{C_2\mathbf{k}\beta}, \tag{32a}$$

$$\hat{\sigma}_v^0 \, \hat{c}_{\mathbf{k}\alpha} \, (\hat{\sigma}_v^0)^{-1} = [-i\sigma_x]_{\alpha\beta} \, \hat{c}_{\sigma_v\mathbf{k}\beta}, \tag{32b}$$

$$\hat{\sigma}_d^0 \, \hat{c}_{\mathbf{k}\alpha} \, (\hat{\sigma}_d^0)^{-1} = \left[-i\hat{\mathbf{n}}' \cdot \sigma\right]_{\alpha\beta} \, \hat{c}_{\sigma_d\mathbf{k}\beta}. \tag{32c}$$

Where $\hat{\mathbf{n}}' = (\hat{\mathbf{x}} - \hat{\mathbf{y}})/\sqrt{2}$ and the Einstein summation convention is implied. With two fold rotations and hence mirrors now squaring to fermion parity, the normal state PSG is

$$(\omega_0(C_2, C_2), \omega_0(\sigma_v, \sigma_v), \omega_0(\sigma_d, \sigma_d)) = (-1, -1, -1). \tag{33}$$

In the absence of spin rotation invariance in the normal state, the pair wavefunction is an admixture of singlet and triplet parts and takes the form

$$\Delta_{\alpha\beta}(\mathbf{k}) = \Psi(\mathbf{k})\left[i\sigma_y\right]_{\alpha\beta} + \mathbf{d}(\mathbf{k}) \cdot \left[\vec{\sigma}(i\sigma_y)\right]_{\alpha\beta}, \tag{34}$$

where Pauli exclusion now requires the three component complex vector $\mathbf{d}$ to obey $\mathbf{d}(\mathbf{k}) = -\mathbf{d}(-\mathbf{k})$. Since the $C_2$ character in all the one dimensional irreps is $+1$, we must have

$$(i\sigma_z)\Delta(\mathbf{k})(i\sigma_z)^T = \Delta(C_2\mathbf{k}) = \Delta(-\mathbf{k}), \tag{35}$$

where the last equality follows from the fact that we are in two spatial dimensions. It is immediately seen that this implies $\mathbf{d}_z(\mathbf{k}) = \mathbf{d}_z(-\mathbf{k})$ and the only way this could be consistent with the constraint imposed by Pauli exclusion is when $\mathbf{d}_z(\mathbf{k}) = 0$. Similarly, by effecting transformations for $\sigma_v$ and $\sigma_d$ on the pairing term we conclude that to tranform as a 1D irrep labeled by the characters $(e^{i\Phi_{\sigma_v}}, e^{i\Phi_{\sigma_d}})$, the non-zero components of the $\mathbf{d}$ vector, must satisfy

$$\left(+d_x(\mathbf{k}), -d_y(\mathbf{k})\right) = e^{i\Phi_{\sigma_v}}\left(d_x(\sigma_v\mathbf{k}), d_y(\sigma_v\mathbf{k})\right), \tag{36a}$$

$$\left(-d_y(\mathbf{k}), -d_x(\mathbf{k})\right) = e^{i\Phi_{\sigma_d}}\left(d_x(\sigma_d\mathbf{k}), d_y(\sigma_d\mathbf{k})\right), \tag{36b}$$

and $\Psi(\mathbf{k})$, like in the case for weak SOC, satisfies (28). Table 2 provides examples of the $\mathbf{d}(\mathbf{k})$ vector for each pairing symmetry. All of these examples belong to a $(p + ip) \uparrow + (p - ip) \downarrow$ type SC. As before, square roots of the characters of the 1D irrep form the compensating phases which modify the transformations in (32) and different SC state PSGs are obtained

Table 2: Tetragonal Symmetry($X = C_{4v}$) with strong SOC. Here $\mathcal{R}_{\text{pair}} \equiv (e^{i\Phi_{\sigma_v}}, e^{i\Phi_{\sigma_d}})$ and $\mathcal{R}_\Phi \equiv (e^{-i\Phi_{C_2}/2}, e^{-i\Phi_{\sigma_v}/2}, e^{-i\Phi_{\sigma_d}/2})$.

| $\mathcal{R}_{\text{pair}}$ | $\mathbf{d(k)}$ | $\mathcal{R}_\Phi$ | $\omega_\Phi$ | $\tilde\omega$ |
|---|---|---|---|---|
| $A_1 : (+1, +1)$ | $k_y\hat{\mathbf{x}} - k_x\hat{\mathbf{y}}$ | $(\pm1, \pm1, \pm1)$ | $(+1, +1, +1)$ | $(-1, -1, -1)$ |
| $A_2 : (-1, -1)$ | $k_x\hat{\mathbf{x}} + k_y\hat{\mathbf{y}}$ | $(\pm1, \pm i, \pm i)$ | $(+1, -1, -1)$ | $(-1, +1, +1)$ |
| $B_1 : (+1, -1)$ | $k_y\hat{\mathbf{x}} + k_x\hat{\mathbf{y}}$ | $(\pm1, \pm1, \pm i)$ | $(+1, +1, -1)$ | $(-1, -1, +1)$ |
| $B_2 : (-1, +1)$ | $k_x\hat{\mathbf{x}} - k_y\hat{\mathbf{y}}$ | $(\pm1, \pm i, \pm1)$ | $(+1, -1, +1)$ | $(-1, +1, -1)$ |

for the four pairing symmetries as outlined in table 2. A few comments are in order. First, comparing the two tables we observe that since the 1D projective representation $\mathcal{R}_\Phi$ formed by the compensating phases and the corresponding cocycle $\omega_\Phi$ depend solely on the pairing symmetry, the correspondence between $\mathcal{R}_{\text{pair}}$ and $\omega_\Phi$ is identical irrespective of the strength of SOC. The difference in the normal state PSG $\omega_0$ accounts for the difference in the SC state PSG $\tilde\omega$ between the corresponding rows of tables 1 and 2.

Second, a question arises as to why only four of the eight PSGs appear in each of the two tables. The answer is apparent once we observe that the $\omega_\Phi$ column only contains the four PSGs with $\omega_\Phi(C_2, C_2) = +1$. This is easily seen as follows. Group law tells us that $\sigma_v\sigma_d = C_2\sigma_d\sigma_v$. Then for any 1D projective representation $\phi$, we must have $\phi(\sigma_v)\phi(\sigma_d) = \pm\phi(C_2)\phi(\sigma_d)\phi(\sigma_v)$. Since $\phi$'s are all non-zero complex numbers, dividing both sides by $\phi(\sigma_v)\phi(\sigma_d)$ gives $\phi(C_2) = \pm1$ and hence $\omega_\phi(C_2, C_2) = +1$. In other words PSGs with $\omega(C_2, C_2) = -1$ cannot have a 1D representation.

Finally, both tables show a one-one correspondence between the four pairing symmetries and four out of the eight possible PSGs. Knowledge of the SC state PSG (from topological or spectroscopic properties) thus uniquely determines the pairing symmetry.

## 4.2 Hexagonal symmetry

Consider a two dimensional honeycomb lattice in the $xy$ plane with a plaquette center chosen as the origin and the $x$-axis passing through a bond center. A six fold rotation about the $z$-axis, $C_6$ and a reflection about a vertical mirror $\sigma_v$ in the $yz$ plane then transform the coordinates as

$$(x, y) \xrightarrow{C_6} \left(\frac{1}{2}x - \frac{\sqrt{3}}{2}y, \frac{1}{2}y + \frac{\sqrt{3}}{2}x\right), \tag{37a}$$

$$(x, y) \xrightarrow{\sigma_v} (-x, y), \tag{37b}$$

$C_6$ and $\sigma_v$ generate the point group $C_{6v}$. It comprises of six rotations and six mirror reflections and the group law is captured by the relations $C_6^6 = e$, $\sigma_v^2 = e$ and $C_6\sigma_v C_6 = \sigma_v$. From these relations it is evident that the $C_6$ and $\sigma_v$ characters in a 1D linear irrep of $C_{6v}$ could only be $\pm1$. Indeed, there are four 1D irreps for this group labeled by $(e^{i\Phi_{C_6}}, e^{i\Phi_{\sigma_v}}) = (\pm1, \pm1)$. Here we note that not only is the group $D_6$ isomorphic to $C_{6v}$, but has indistinguishable action in two spatial dimensions. In $D_6$, the two-fold rotation about the in-plane $y$-axis, $C_{2y}$ assumes the role of $\sigma_v$ in $C_{6v}$. Thus, when we are strictly in two spatial dimensions, $C_{6v}$ and $D_6$ can be used interchangeably.

Since $\mathcal{H}^{(2)}(C_{6v}, \mathbb{Z}_2) = \mathbb{Z}_2^3$, there are eight possible PSGs distinguished on the basis of whether $C_2$ and $\sigma_v$ square to $+1$ or $-1$ and whether they commute or anti-commute.

The classes of 2-cocycles are labeled by

$$\left( \omega(C_2, C_2), \omega(\sigma_v, \sigma_v), \frac{\omega(C_2, \sigma_v)}{\omega(\sigma_v, C_2)} \right) = (\pm 1, \pm 1, \pm 1). \tag{38}$$

We discuss the case when the normal and the SC states have spin rotation invariance. Denoted by the superscript 0, the transformations that preserve the kinetic energy are

$$\hat{C}_6^0 \hat{c}_{\mathbf{k}\alpha s}(\hat{C}_6^0)^{-1} = (\tau_x)_{\alpha\beta} \, \hat{c}_{C_6 \mathbf{k}\beta s}, \tag{39a}$$

$$\hat{\sigma}_v^0 \hat{c}_{\mathbf{k}\alpha s}(\hat{\sigma}_v^0)^{-1} = \hat{c}_{\sigma_v \mathbf{k}\alpha s}. \tag{39b}$$

Where $\alpha, \beta$ are sub-lattice labels, $s$ labels spin and $\vec{\tau}$ denotes Pauli matrices in the sub-lattice space. The momentum $\mathbf{k}$ is measured from the $\Gamma$ point of the Brilloin zone. The normal state PSG is trivial with

$$\left( \omega_0(C_2, C_2), \omega_0(\sigma_v, \sigma_v), \frac{\omega_0(C_2, \sigma_v)}{\omega_0(\sigma_v, C_2)} \right) = (+1, +1, +1). \tag{40}$$

Here we consider a generic situation where both the bands participate in pairing and we express the pair wavefunction in the sub-lattice basis. If however we have a weak coupling scenario in which only a single band takes part in pairing, it is more convenient to express the pair wavefunction in the active band basis. For the present case, consistent with Pauli exclusion, the spin singlet wave function has the form

$$[\Delta(\mathbf{k})]_{\alpha s \beta s'} = \Psi_{\alpha\beta}(\mathbf{k})(i\sigma_y)_{ss'}, \tag{41}$$

where $\Psi_{\alpha\beta}(\mathbf{k}) = \Psi_{\beta\alpha}(-\mathbf{k})$. For the pairing term to transform as the irrep $(e^{i\Phi_{C_6}}, e^{i\Phi_{\sigma_v}})$ under (39), $\Psi_{\alpha\beta}(\mathbf{k})$ satisfies

$$(\tau_x)_{\alpha\gamma} \Psi_{\gamma\delta}(\mathbf{k})(\tau_x)_{\beta\delta} = e^{i\Phi_{C_6}} \Psi_{\alpha\beta}(C_6\mathbf{k}), \tag{42a}$$

$$\Psi_{\alpha\beta}(\mathbf{k}) = e^{i\Phi_{\sigma_v}} \Psi_{\alpha\beta}(\sigma_v\mathbf{k}). \tag{42b}$$

In table 3 we provide examples of $\Psi_{\alpha\beta}(\mathbf{k})$ satisfying (42) for each pairing symmetry. The compensating phases $(e^{-i\Phi_{C_6}/2}, e^{-i\Phi_{\sigma_v}/2})$ forming the 1D projective representation $\mathcal{R}_\Phi$ and the corresponding 2-cocycle $\omega_\Phi$ are also tabulated. A product of $\omega_\Phi$ and $\omega_0$ then gives the SC state PSG $\tilde{\omega}$. The four pairing symmetries correspond to four distinct $\tilde{\omega}$ s. The SC state PSG thus uniquely determines the pairing symmetry for this point group. Like in the previous case, only four out of the eight possible PSGs appear in table 3. Inspecting the $\omega_\Phi$ column we observe that it only has entries with $\omega_\Phi(C_2, \sigma_v)/\omega_\Phi(\sigma_v, C_2) = +1$. Since complex numbers always commute, it is impossible to get a 1D projective representation of $C_{6v}$ where $\omega_\Phi(C_2, \sigma_v)/\omega_\Phi(\sigma_v, C_2) = -1$.

We end this subsection discussing superconductivity in magic angle twisted bilayer graphene (MATBG) where the pairing symmetry is still not known although there has been some theoretical proposals [25]. The experimental observation of nematicity in the SC state [26], shows that the normal state $D_6$ symmetry, is spontaneously broken in the SC state. Thus condensation must take place either in the $E_1$ or the $E_2$ irrep of $D_6$. As pointed out in the introduction, if it were any of the 1D irreps, the pair wavefunction would be invariant under $D_6$ up-to a phase rotation, and the SC state would not show the observed nematicity. This corresponds to the orbital part being a $p$-wave for the $E_1$ irrep or a $d$-wave for the $E_2$ irrep in the pair wavefunction proposed in [25]. The residual symmetry in the SC state is the two-fold rotation about $z$-axis, $X = C_{2z}$. Since the $E_1$ irrep ($p$-wave) has a $C_2$ character $-1$ and the $E_2$ irrep ($d$-wave) has a $C_2$ character $+1$, these correspond to the two 1D irreps of $X$. There is a one to one correspondence between $\mathcal{R}_{\text{pair}}$ and PSGs for $X$ as shown in table B and thus the two possible pairing symmetries would give two distinct SC state PSGs.

Table 3: Hexagonal symmetry ($X = C_{6v}$) with weak SOC. Here $\mathcal{R}_{pair} = (e^{i\Phi_{C_6}}, e^{i\Phi_{\sigma_v}})$, $\mathcal{R}_\Phi = (e^{-i\Phi_{C_6}/2}, e^{-i\Phi_{\sigma_v}/2})$. Also, $f(\mathbf{k}) = k_x k_y (k_x^2 - 3k_y^2)(k_y^2 - 3k_x^2)$ and $g(\mathbf{k}) = k_x(3k_y^2 - k_x^2)$.

| $\mathcal{R}_{pair}$ | $\Psi_{AA}(\mathbf{k})$ | $\Psi_{BB}(\mathbf{k})$ | $\Psi_{AB}(\mathbf{k})$ | $\mathcal{R}_\Phi$ | $\omega_\Phi$ | $\tilde{\omega}$ |
|---|---|---|---|---|---|---|
| $A_1 = (+1, +1)$ | $\Delta_0$ | $\Delta_0$ | $\Delta_0'$ | $(\pm 1, \pm 1)$ | $(+1, +1, +1)$ | $(+1, +1, +1)$ |
| $A_2 = (+1, -1)$ | $\Delta_0 f(\mathbf{k})$ | $\Delta_0 f(\mathbf{k})$ | $\Delta_0' g(\mathbf{k})$ | $(\pm 1, \pm i)$ | $(+1, -1, +1)$ | $(+1, -1, +1)$ |
| $B_1 = (-1, +1)$ | $\Delta_0$ | $-\Delta_0$ | $0$ | $(\pm i, \pm 1)$ | $(-1, +1, +1)$ | $(-1, +1, +1)$ |
| $B_2 = (-1, -1)$ | $\Delta_0 f(\mathbf{k})$ | $-\Delta_0 f(\mathbf{k})$ | $0$ | $(\pm i, \pm i)$ | $(-1, -1, +1)$ | $(-1, -1, +1)$ |

### 4.3 Trigonal symmetry

Like in the previous subsection, we consider the honeycomb lattice in the $xy$ plane except now two different species occupy the $A$ and $B$ sub-lattices. Such is the case, for example, in a mono-layer transition metal dichalcogenide (TMD). The resulting point group $C_{3v}$ is generated by a three-fold rotation about the $z$-axis ($C_3$) and reflection about a vertical mirror in the $yz$ plane ($\sigma_v$) which act on the coordinates as

$$(x, y) \xrightarrow{C_3} \left( -\frac{1}{2} x - \frac{\sqrt{3}}{2} y, -\frac{1}{2} y + \frac{\sqrt{3}}{2} x \right), \tag{43a}$$

$$(x, y) \xrightarrow{\sigma_v} (-x, y). \tag{43b}$$

The relations $C_3^3 = \sigma_v^2 = e$ and $C_3 \sigma_v C_3 = \sigma_v$ capture the group law. There are two 1D irreps for this group with $e^{i\Phi_{\sigma_v}} = \pm 1$ and two PSGs with $\sigma_v$ squaring to unity in one and to the fermion parity in the other, $\omega(\sigma_v, \sigma_v) = \pm 1$.

In TMDs, the presence of strong Ising SOC breaks spin rotation invariance [27]. Hole doping away from charge neutrality creates small Fermi surface pockets at the $K$ and $K'$ valleys. Denoted by superscript 0, the symmetry operations that preserve the kinetic energy act on the fermion operator $\hat{c}_{\mathbf{k}\nu s}$ for the active band as

$$\hat{C}_3^0 \, \hat{c}_{\mathbf{k}\nu s} \, (\hat{C}_3^0)^{-1} = \left[ e^{-i\frac{\pi}{3}\sigma_z} \right]_{ss'} \hat{c}_{C_3\mathbf{k}\nu s'}, \tag{44a}$$

$$\hat{\sigma}_v^0 \, \hat{c}_{\mathbf{k}\nu s} \, (\hat{\sigma}_v^0)^{-1} = [\tau_x]_{\nu\nu'} [i\sigma_x]_{ss'} \hat{c}_{\sigma_v\mathbf{k}\nu's'}, \tag{44b}$$

where $\nu$ is the valley and $s$ is the spin label and momentum $\mathbf{k}$ is measured from the $K$ or $K'$ point. Pauli matrices $\vec{\sigma}$ and $\vec{\tau}$ act on spin and valley spaces respectively. The normal state PSG is thus described by the cocycle $\omega_0(\sigma_v, \sigma_v) = -1$.

To ensure the Cooper pair has a zero center of mass momentum, pairing must be inter-valley. Because of time reversal invariance, the Fermi surface pockets at opposite valleys have oppositely polarized spins. If the spin polarization is $\sigma_z = +1$ in the $K$ valley ($\tau_z = +1$), then it is along $\sigma_z = -1$ in the $K'$ valley ($\tau_z = -1$). Therefore, in addition to Pauli exclusion, the order parameter matrix $\Delta(\mathbf{k})$ in the spin-valley space must satisfy the constraint $\mathcal{P}^T \Delta(\mathbf{k}) = \Delta(\mathbf{k})\mathcal{P} = \Delta(\mathbf{k})$ where $\mathcal{P} = \frac{1}{2}(\mathbb{1} + \sigma_z \tau_z)$ projects onto the $\sigma_z \tau_z = +1$ space. Consistent with these requirements, $\Delta(\mathbf{k})$ takes the form

$$\Delta(\mathbf{k}) = \left( \Psi(\mathbf{k})\tau_+ - \Psi(-\mathbf{k})\tau_- \right)(\hat{\mathbf{z}} \cdot \vec{\sigma})(i\sigma_y) + \left( \Psi(\mathbf{k})\tau_+ + \Psi(-\mathbf{k})\tau_- \right)(i\sigma_y). \tag{45}$$

Table 4: Trigonal symmetry ($X = C_{3v}$) with strong SOC.

| $\mathcal{R}_{pair} \equiv e^{i\Phi_{\sigma_v}}$ | $\Psi(\mathbf{k})$ | $\mathcal{R}_\Phi \equiv e^{-i\Phi_{\sigma_v}/2}$ | $\omega_\Phi$ | $\tilde{\omega}$ |
|---|---|---|---|---|
| $A_1 = +1$ | $\Delta_0$ | $\pm 1$ | $+1$ | $-1$ |
| $A_2 = -1$ | $\Delta_0 k_y(3k_x^2 - k_y^2)$ | $\pm i$ | $-1$ | $+1$ |

As expected, the absence of spin rotation invariance in the normal state results in a pair wavefunction which is a superposition of singlet and triplet parts. For the pairing term to transform as a 1D irrep of $C_{3v}$ under (44), $\Psi(\mathbf{k})$ must satisfy

$$\Psi(\mathbf{k}) = \Psi(C_3\mathbf{k}), \tag{46a}$$

$$\Psi(-\mathbf{k}) = e^{i\Phi_{\sigma_v}}\Psi(\sigma_v\mathbf{k}). \tag{46b}$$

Table 4 shows that the two 1D irreps are in a one-one correspondence with the two SC state PSGs. It also gives an example of $\Psi(\mathbf{k})$ for each pairing symmetry.

## 4.4 Physical consequences of the PSG

The projective symmetry group $G_f$ of the BdG Hamiltonian has effects on all fermionic excitations of the superconductor, since the Bogoliubov quasipaticles as excitations of the BdG Hamiltonian form a linear representation of the PSG $G_f$. In particular, the topological properties of the superconductor is determined by the PSG, as different PSGs give rise to different classifications of fermion topological superconductors (TSCs) [11,14,28]. This is a well-known fact in the classification of gapped fermion topological phases, both in the 10-fold way [29] classification of non-interacting topological superconductors [11,30], and in the interacting classification of fermion symmetry protected topological phases [14,31]. For example, in the case of time reversal symmetry $\mathcal{T}$, it is well known that two- and three-dimensional topological insulators only exist for spinful electrons with $\hat{\mathcal{T}}^2 = (-1)^{\hat{F}}$ and $G_f = U(1) \rtimes Z_4^{\mathcal{T}}$, which is a different symmetry class (class AII in the 10-fold way [29]) than spinless case (class AI in the 10-fold way [29]), with $\hat{\mathcal{T}}^2 = 1$ and $G_f = U(1) \rtimes Z_2^{\mathcal{T}}$. In addition to topological classifications, these two distinct symmetry classes have many other different properties, such as the presence vs. absence of Kramers degeneracy of fermion excitations. Below we illustrate how different PSGs, and hence different pairing symmetries, give rise to different classifications of TSCs, in the case of crystalline symmetries [28,32–37]. Our classification scheme applies to gapped topological SCs. Thus, weak pairing unconventional SC with gap nodes are not part of the classification. However, there are fully gapped unconventional SCs, like the $(p + ip)$ state in 2D and the B-phase of He3, which are topologically non-trivial. Our analysis focus on understanding how, in the presence of additional crystalline symmetries, pairing symmetry through the PSG affects the classification of such states. We use 3d SCs with mirror reflection symmetry $M_x$, and 2d SCs with 2-fold rotational symmetry $C_{2z}$ as two known examples to demonstrate this fact.

### 4.4.1 3d SCs with mirror reflection symmetry $M_x$

Our first example is the classification of TSCs in three dimension (3d) in the presence of only mirror reflection symmetry $M_x$ which reverses the $x$ coordinate. From the group cohomology $\mathcal{H}^{(2)}(\mathbb{Z}_2^{M_x}, \mathbb{Z}_2) = \mathbb{Z}_2$, we find two possible fermion PSGs in the presence of strong SOCs: $G_f = \mathbb{Z}_2^{\hat{M}_x} \times \mathbb{Z}_2^F$ with $\hat{M}_x^2 = +1$, and $G_f = \mathbb{Z}_4^{\hat{M}_x}$ with $\hat{M}_x^2 = (-1)^{\hat{F}}$. Similarly, in the presence of weak SOCs and spin rotational symmetry, the two possible PSGs are given by $G_f = SU(2) \times \mathbb{Z}_2^{\hat{M}_x}$ with $\hat{M}_x^2 = +1$, and $G_f = SU(2) \times Z_4^{\hat{M}_x}/\mathbb{Z}_2$ with $\hat{M}_x^2 = (-1)^{\hat{F}}$.

Table 5: Classification of class D topological superconductor in 3d with mirror reflection $M_x$. The fermion projective symmetry groups $G_f$ are listed for superconductors with weak/strong SOCs and $A'/A''$ pairing symmetries. Note that the topological classification is solely determined by $G_f$.

| SOC strength | pairing symmetry | $G_f$ | K-theory classification [38, 39] |
|---|---|---|---|
| Weak | $A'$ | $SU(2) \times \mathbb{Z}_2^{\hat{M}_x}$ | $\mathbb{Z}$ |
| | $A''$ | $SU(2) \times \mathbb{Z}_4^{\hat{M}_x}/\mathbb{Z}_2$ | $\mathbb{Z}_2$ |
| Strong | $A'$ | $\mathbb{Z}_4^{\hat{M}_x}$ | 0 |
| | $A''$ | $\mathbb{Z}_2^{\hat{M}_x} \times \mathbb{Z}_2^F$ | $\mathbb{Z}$ |

Table 6: Classification of class D topological superconductor in 2d with $C_{2z}$ rotation perpendicular to the 2d $x$-$y$ plane. The fermion PSGs $G_f$ are listed for superconductors with weak/strong SOC and $A/B$ pairing symmetries. Note that the topological classification is solely determined by $G_f$.

| SOC strength | pairing symmetry | $G_f$ | K-theory classification [40, 41] |
|---|---|---|---|
| Weak | $A$ | $SU(2) \times \mathbb{Z}_2^{\tilde{C}_{2z}}$ | $\mathbb{Z}$ |
| | $B$ | $SU(2) \times \mathbb{Z}_4^{C_{2z}}/\mathbb{Z}_2$ | $\mathbb{Z}^2$ |
| Strong | $A$ | $\mathbb{Z}_4^{\tilde{C}_{2z}}$ | $\mathbb{Z}^2$ |
| | $B$ | $\mathbb{Z}_2^{C_{2z}} \times \mathbb{Z}_2^F$ | $\mathbb{Z}$ |

For weakly interacting systems, $K$-theory [10, 11, 30, 38, 39] can be used to classify distinct TSCs described by BdG Hamiltonians. In the presence of strong SOCs, it gives rise to a $\mathbb{Z}$ classification of TSCs for the case of $\hat{M}_x^2 = +1$, and a trivial classification for the case of $\hat{M}_x^2 = (-1)^{\hat{F}}$ [38, 39]. In the presence of a weak SOC and $SU(2)$ spin rotational symmetry, there is a $\mathbb{Z}$ classification of TSCs for the case of $\hat{M}_x^2 = +1$, and a $\mathbb{Z}_2$ classification for the case of $\hat{M}_x^2 = (-1)^{\hat{F}}$ [38, 39]. With this result we can now readily bridge the gap between pairing symmetry and the K-theory classification of TSC via the projective symmetry group $G_f$.

A mirror symmetry satisfying $\hat{M}_x^2 = +1$ is preserved either in a singlet superconductor with pairing symmetry $A'$ in the presence of a weak SOC, or in a superconductor with pairing symmetry $A''$ in the presence of a strong SOC. The classifications of weakly-interacting TSCs in these two cases are both $\mathbb{Z}$.

To compare, a mirror symmetry satisfying $\hat{M}_x^2 = (-1)^{\hat{F}}$ corresponds to either a singlet superconductor with pairing symmetry $A''$ in the presence of a weak SOC, or a pairing symmetry $A'$ in the presence of a strong SOC. For these two cases the classifications of TSCs are $\mathbb{Z}_2$ and trivial, respectively. The results are summarized in Table 5.

### 4.4.2 2d SCs with 2-fold rotational symmetry $C_{2z}$

Our second example is the classification of TSCs in two dimensions (2d) with a $C_{2z}$ rotation perpendicular to the 2d plane. In this case $\mathcal{H}^{(2)}(C_{2z}, \mathbb{Z}_2) = \mathbb{Z}_2$, which yields two different fermion PSGs in the presence of a strong (weak) SOC: one with $\hat{C}_{2z}^2 = +1$ and the other with $\hat{C}_{2z}^2 = (-1)^{\hat{F}}$, as shown in Table 6. Accordingly, the $K$-theory classification of $C_{2z}$ symmetric TSCs [40] are given by $\mathbb{Z}$ for $\hat{C}_{2z}^2 = +1$ and $\mathbb{Z}^2$ for $\hat{C}_{2z}^2 = (-1)^{\hat{F}}$.

From the relationship between pairing symmetry and projective symmetry group, we find that the $\hat{C}_{2z}^2 = +1$ case corresponds to either a singlet SC with pairing symmetry $A$ or a SC with a strong SOC and pairing symmetry $B$. Then for these two cases the classifications of topological superconductors are both $\mathbb{Z}$.

The $\hat{C}_{2z}^2 = (-1)^{\hat{F}}$ case corresponds to either a singlet superconductor with pairing symmetry $B$ or a superconductor with a strong SOC and pairing symmetry $A$. For these two cases the classifications of TSCs are both $\mathbb{Z}^2$. The results are summarized in Table 6.

From these two examples, we see that BdG Hamiltonians with different PSGs generally give rise to different topological classifications. Based on the correspondence between the fermion PSG and the pairing symmetry discussed in sections 2-3, the classification of TSCs is therefore directly related to the pairing symmetry, as demonstrated in Table 5-6. For TSCs of all the possible pairing symmetries associated with a magnetic point group symmetry, Ref. [28] summarizes a full list of K-theory classification for both the cases of spinless (weak SOC) and spinful (strong SOC) electrons.

# 5 General framework

So far we have only focused on cases where the normal and the SC states have the same spin rotational symmetry. This simplifies the the form of the fermion PSG as explained below. For systems with weak SOC the physical symmetry group is $G = X \times SO(3)_{\text{spin}}$. When we take a central extension by the fermion parity group to obtain the fermion PSG, both in the normal and SC state PSGs, the $SO(3)_{\text{spin}}$ becomes an $SU(2)_{\text{spin}}$. The spatial part however undergoes different central extensions: $X_f^0$ in the normal state PSG and $\tilde{X}_f$ in the SC state PSG. Thus, the fermion PSG preserving the kinetic energy is $(X_f^0 \times SU(2))/\mathbb{Z}_2$ and that preserving the BdG is $(\tilde{X}_f \times SU(2))/\mathbb{Z}_2$ (taking a quotent by $\mathbb{Z}_2$ takes care of the "double-counting" of $\mathbb{Z}_2^F$). Thus, the difference between the normal and SC state PSGs is completely captured by different *central* extensions of $X$ by $\mathbb{Z}_2^F$. This holds true for systems with strong SOC where spin rotation symmetry is altogether absent and with $G = X$, the fermion PSG is synonymous with the central extension of $X$ by $\mathbb{Z}_2^F$.

When spin rotation is spontaneously broken in the SC state, the fermion PSG no longer admits such a simple description in terms of central extensions of the spatial part. When the physical symmetry group in the SC state is $G = X \times S$ where $S$ is a subgroup of the normal state spin rotation group, as we show in section 5.1, the fermion PSG could be a generic *group* extension of $X$ by the fermion spin rotation symmetry group $S_f$. $S_f$ in turn is a central extension of $S$ by the fermion parity group and could in general be non-Abelian.

In superfluid $^3$He, condensation into the spin triplet channel spontaneously breaks the continuous spin rotation symmetry present in the normal state. We discuss it in section 5.2 in the light of this general framework.

## 5.1 Group extension and pairing symmetry in a generic superconductor

Let the normal state spin rotational symmetry group $S_0 \subseteq SO(3)_{\text{spin}}$ be spontaneously broken down to $S \subseteq S_0$ in the SC state. With the charge $U(1)$ symmetry in the normal state completely broken, the SC state physical (bosonic) symmetry group $G$ takes the form $G = X \times S$, where $X$ denotes the spatial symmetry group preserved by the SC state. We now describe the structure of the fermion symmetry group $G_f$ in such cases. Some of the relevant mathematical details can be found in Appendix A.

First of all, the fermion spin rotation (or internal) symmetry group in the SC-state, $S_f$ is a subgroup of $G_f$ and given by a *central* extension of the physical spin rotation symmetry group $S$:

$$1 \to \mathbb{Z}_2^F \to S_f \to S \to 1 \,. \tag{47}$$

$S_f$ has the form $S_f = \left\{ (\pm 1)^{\hat{F}} \hat{s}' \,|\, s \in S \right\}$ where under a spin rotation $\hat{s}'$, the fermion operator transforms as

$$\hat{s}' \, \hat{c}_{\mathbf{k}\alpha} \, \hat{s}'^{-1} = \left[ \tilde{U}^s \right]_{\alpha\beta}^{\dagger} \, \hat{c}_{\mathbf{k}\beta} \,, \tag{48a}$$

$$\tilde{U}^s = e^{-i\phi_s} \, U_0^s \,. \tag{48b}$$

The transformation is a combination of an $SU(2)$ spin rotation $U_0^s$ that preserves the kinetic energy and a compensating phase rotation $e^{-i\phi_s}$ required to make $\hat{s}'$ a symmetry of the BdG hamiltonian. Being an internal (on-site) symmetry, $\hat{s}'$ leaves the momentum label unchanged on both sides of (48a). For a given $S$, the possible choices for $S_f$ is captured by $\mathcal{H}^{(2)}(S, \mathbb{Z}_2)$, the second cohomology group formed by inequivalent classes of cocycles $[\tilde{\omega}]$. As already noted in previous sections, the cocycle $\tilde{\omega}$ taking values in $\{\pm 1\}$ also characterize the projective representation of $S$ formed by $\left\{ \tilde{U}^s \,|\, s \in S \right\}$.

To build $G_f$, next we need to consider the group of spatial symmetries, $X$. For $g \in X$, $\hat{g}_0$ preserves the kinetic energy and transforms the fermion operator as

$$\hat{g}_0 \, \hat{c}_{\mathbf{k}\alpha} \, \hat{g}_0^{-1} = \left[ U_0^g(\mathbf{k}) \right]_{\alpha\beta}^{\dagger} \, \hat{c}_{g\mathbf{k}\beta} \,. \tag{49}$$

To make this a symmetry of the pairing term, not only do we need to dress it with a compensating phase $e^{-i\phi_g}$ but also with a normal state spin rotation $U_0^{s_0(g)}$ (where $s_0(g) \in S_0$). Since the kinetic energy is invariant under normal state spin rotations and $U(1)$ phase rotations, the resulting transformation $\hat{g}'$ preserves the BdG hamiltonian. Its action on the fermion operator is given by

$$\hat{g}' \, \hat{c}_{\mathbf{k}\alpha} \, \hat{g}'^{-1} = \left[ \tilde{U}^g(\mathbf{k}) \right]_{\alpha\beta}^{\dagger} \, \hat{c}_{g\mathbf{k}\beta} \,, \tag{50a}$$

$$\tilde{U}^g(\mathbf{k}) = e^{-i\phi_g} \, U_0^{s_0(g)} \, U_0^g(\mathbf{k}) \,. \tag{50b}$$

Although the structure of $G_f$ is in general much more complicated than simply a direct (or even a semi-direct) product of spatial and spin rotation symmetry groups, it is possible to obtain a generic characterization as discussed below. To begin with, let us compare what one obtains by the successive application of $\hat{h}'$ and $\hat{g}'$ on $\hat{c}_{\mathbf{k}\alpha}$ and that by applying $\widehat{gh}'$ on the same. Using (50a) we see that in both these cases we get a fermion operator on the right hand side with the same momentum label $gh\mathbf{k}$. With both $\hat{g}'\hat{h}'$ and $\widehat{gh}'$ being symmetries of the BdG hamiltonian, this implies that these are in fact the same up to an internal symmetry transformation $(\eta)^{\hat{F}} \hat{s}'(g, h)$. In other words,

$$\hat{g}' \hat{h}' = (\eta)^{\hat{F}} \hat{s}'(g, h) \, \widehat{gh}' \,. \tag{51}$$

Moreover, for any $\hat{s}' \in S_f$, the transformation $\hat{g}' \hat{s}' \hat{g}'^{-1}$ keeps the momentum label of the fermion operator unchanged and hence must belong to $S_f$. Then again, any element of $G_f$ can be written as a product of a $\hat{g}'$ for some $g \in X$ and an $(\eta)^{\hat{F}} \hat{s}' \in S_f$ such that $G_f$ has the form $G_f = \left\{ (\pm 1)^{\hat{F}} \hat{s}' \hat{g}' \,|\, s \in S, g \in X \right\}$. We thus conclude that $S_f$ is a normal subgroup of $G_f$ and $G_f / S_f = X$. Equivalently $S_f$, $G_f$ and $X$ satisfy the short exact sequence

$$1 \to S_f \to G_f \to X \to 1 \,. \tag{52}$$

It is hard to find all such extensions in the most general case. However, if $S_f$ is abelian then all such extensions are captured by the second cohomology group $\mathcal{H}^{(2)}_{[\rho]}(X, S_f)$. It is to be noted that the matrices $\{\tilde{U}^s \cdot \tilde{U}^g | s \in S, g \in X\}$ form a projective representation of $G = S \times X$ with coefficients in $\{\pm \tilde{U}^s | s \in S\}$.

Finally, we discuss the relation between the fermion PSG $G_f$ and the pairing symmetry. In general, the pairing wavefunctions $\Delta_{\alpha, \beta}$ in BdG Hamiltonian (7) form a linear representation $\mathcal{R}_{pair}$ of the bosonic symmetry group $G = S \times X$, where $S$ stands for the global (spin rotational) symmetry group and $X$ stands for the crystalline symmetry group. Meanwhile, in the representation $\{\tilde{U}^s \cdot \tilde{U}^g | s \in S, g \in X\}$ introduced above, we can identify a projective representation of group $G = S \times X$:

$$\mathcal{R}_\Phi(s, g) = e^{-\mathrm{i}(\phi_s + \phi_g)} U_0^{s_0(g)}, \qquad \forall\, s \in S, \quad g \in X. \tag{53}$$

It is evident that the projective representation $\mathcal{R}_\Phi$ is not 1D in general. Also note that while for the global internal symmetry group $S$, the transformations that preserve the kinetic energy, preserve the pairing term up to a phase (just as in (14)), that may not be the case for the crystalline group $X$. Hence $\mathcal{R}_{\text{pair}}$ is in general a multi-dimensional linear representation of $G$. $\mathcal{R}_{\text{pair}}$ and $\mathcal{R}_\Phi$ are related by the following relation:

$$\mathcal{R}_\Phi \otimes \mathcal{R}_\Phi \otimes \mathcal{R}_{pair} = \mathbb{1} \oplus \cdots, \tag{54}$$

where $\mathbb{1}$ denotes the trivial one-dimensional (1d) representation of group $G$. This is because the pairing term (9) must remain invariant under the PSG symmetry transformation $\{\tilde{U}^s \cdot \tilde{U}^g | s \in S, g \in X\}$. Notice that in the special case of $\mathcal{R}_\Phi$ being a 1d irrep, applicable to the situation discussed in Section 3, the general relation (54) reduces to Eq. (22). In order for $\mathcal{R}_{pair}$ to be a multi-dimensional irrep., the tensor product $\mathcal{R}_\Phi \otimes \mathcal{R}_\Phi$ of two projective representations in Eq. (54) must be a multi-dimensional irrep. of group $G$. Therefore, a necessary condition for the pairing order parameter to form a multi-dimensional irrep. of symmetry group $G$ (i.e. for $\mathcal{R}_{pair}$ to be multi-dimensional) is that $\mathcal{R}_\Phi$ is a multi-dimensional projective representation of group $G$. As we will show below, one such example is the superfluid B phase of Helium 3.

## 5.2  Examples: Superfluid A and B phases in Helium-3

The most famous example of triplet superconductivity (or superfluidity) is perhaps Helium-3 [5]. The normal state preserves continuous spatial rotations and inversion symmetry:

$$X_0 = SO(3)_{\text{orbital}} \times \mathbb{Z}_2^I \simeq O(3), \tag{55}$$

along with full spin rotation symmetry, $S_0 = SO(3)_{\text{spin}}$. Condensation takes place in a spin triplet $p$-wave state breaking the full spin rotation symmetry down to a proper subgroup. In the basis, $\Psi_\mathbf{k} \equiv (c_{\mathbf{k},\uparrow}, c_{\mathbf{k},\downarrow}, c^\dagger_{-\mathbf{k},\uparrow}, c^\dagger_{-\mathbf{k},\downarrow})^T$ the BdG Hamiltonian takes the form

$$\hat{H}_{\text{BdG}} = \sum_\mathbf{k} \hat{\Psi}^\dagger_\mathbf{k} \begin{pmatrix} (\frac{k^2}{2m} - \mu)\mathbb{1} & \Delta(\mathbf{k}) \\ \Delta^\dagger(\mathbf{k}) & (\mu - \frac{k^2}{2m})\mathbb{1} \end{pmatrix} \hat{\Psi}_\mathbf{k}, \tag{56a}$$

$$\Delta(\mathbf{k}) = \mathbf{d}(\mathbf{k}) \cdot \vec{\sigma}(i\sigma_y). \tag{56b}$$

To obey Fermi statistics, the three component complex vector $\mathbf{d}(\mathbf{k})$ must satisfy $\mathbf{d}(\mathbf{k}) = -\mathbf{d}(-\mathbf{k})$. In particular for $p$-wave $^3$He, the components of $\mathbf{d}(\mathbf{k})$ are linear in $\mathbf{k}$. The various phases, characterized by different broken symmetries, are distinguished by the form of the $\mathbf{d}(\mathbf{k})$ vector. We apply the general framework described above to the two phases: (1) B phase, also known

as the Balian-Werthamer (BW) phase [42], (2) A phase, also known as Anderson-Brinkman-Morel (ABM) phase [43,44], discussing the residual symmetry group in the SC state and the SC state fermion PSG in each case.

The transformations that preserve the kinetic energy act on the fermion operators as

$$\text{Spin rot.} \quad \hat{\mathcal{S}}_0(\vec{\theta})\hat{c}_{\mathbf{k}s}\hat{\mathcal{S}}_0(\vec{\theta})^{-1} = \left[U_0^{\mathcal{S}(\vec{\theta})}\right]^{\dagger}_{ss'}\hat{c}_{\mathbf{k}s'}, \qquad U_0^{\mathcal{S}(\vec{\theta})} = e^{i\vec{\theta}\cdot\vec{\sigma}/2}, \qquad (57a)$$

$$\text{Space rot.} \quad \hat{\mathcal{R}}_0(\vec{\theta})\hat{c}_{\mathbf{k}s}\hat{\mathcal{R}}_0(\vec{\theta})^{-1} = \left[U_0^{\mathcal{R}(\vec{\theta})}\right]^{\dagger}_{ss'}\hat{c}_{\mathcal{R}(\vec{\theta})\mathbf{k}s'}, \qquad U_0^{\mathcal{R}(\vec{\theta})} = \mathbb{1}, \qquad (57b)$$

$$\text{Inversion} \quad \hat{\mathcal{I}}_0\hat{c}_{\mathbf{k}s}\hat{\mathcal{I}}_0^{-1} = \left[U_0^{\mathcal{I}}\right]^{\dagger}_{ss'}\hat{c}_{-\mathbf{k}s'}, \qquad U_0^{\mathcal{I}} = \mathbb{1}. \qquad (57c)$$

With $\left[\hat{\mathcal{R}}_0(\pi\hat{\mathbf{n}})\right]^2 = \hat{\mathcal{I}}_0^2 = 1$ and $\left[\hat{\mathcal{S}}_0(\pi\hat{\mathbf{n}})\right]^2 = (-1)^{\hat{F}}$, the normal state fermion PSG is of the form $X_0 \times SU(2)$.

### 5.2.1 Superfluid B phase of Helium-3

In the B phase, $\mathbf{d}(\mathbf{k}) = \Delta_0(k_x\hat{\mathbf{x}} + k_y\hat{\mathbf{y}} + k_z\hat{\mathbf{z}})$ [22]. The spin rotation group is broken down from $S_0 = SO(3)_{\text{spin}}$ to its trivial subgroup $S = \{\mathbb{1}\}$ in the SC state. Using (47), the fermion onsite symmetry group is simply the fermion parity group $S_f = \mathbb{Z}_2^F$. The system remains isotropic in the SC state and $X = SO(3)_{\text{orb.+spin}} \times \mathbb{Z}_2^I \simeq O(3)$. As suggested by the label, the normal state spatial rotation in (57b) has to be modified by including a normal state spin rotation and since $\mathbf{d}(\mathbf{k})$ is inversion odd, the normal state inversion in (57c) has to be modified by a compensating phase rotation by $i$. The transformations that preserve (56a) are

$$\text{Space rot.} \quad \hat{\mathcal{R}}'(\vec{\theta})\hat{c}_{\mathbf{k}s}\hat{\mathcal{R}}'(\vec{\theta})^{-1} = \left[\tilde{U}^{\mathcal{R}(\vec{\theta})}\right]^{\dagger}_{ss'}\hat{c}_{\mathcal{R}(\vec{\theta})\mathbf{k}s'}, \qquad \tilde{U}^{\mathcal{R}(\vec{\theta})} = e^{i\vec{\theta}\cdot\vec{\sigma}/2}\cdot\mathbb{1}, \qquad (58a)$$

$$\text{Inversion} \quad \hat{\mathcal{I}}'\hat{c}_{\mathbf{k}s}\hat{\mathcal{I}}'^{-1} = \left[\tilde{U}^{\mathcal{I}}\right]^{\dagger}_{ss'}\hat{c}_{-\mathbf{k}s'}, \qquad \tilde{U}^{\mathcal{I}} = i\mathbb{1}. \qquad (58b)$$

With $S_f = \mathbb{Z}_2^F$, the full fermion symmetry group $G_f$, which by (52) is a group extension of $X$ by $S_f$, reduces to a central extension of $X$ by $\mathbb{Z}_2^F$. Eqns. (58a) and (58b) give $\hat{\mathcal{R}}'(\pi\hat{\mathbf{n}})^2 = \hat{\mathcal{I}}'^2 = (-1)^{\hat{F}}$, showing that $G_f$ involves non-trivial central extensions of both $SO(3)_{\text{orb.+spin}}$ and $\mathbb{Z}_2^I$ and is given by $G_f = (SU(2) \times \mathbb{Z}_4)/\mathbb{Z}_2$.

Conversely, one can learn about the pair wavefunction from the SC state PSG in the superfluid B phase. From (58a), we see that $\mathcal{R}_\Phi(\mathcal{R}(\vec{\theta})) = U_0^{\mathcal{S}(\vec{\theta})} = e^{i\vec{\theta}\cdot\sigma/2}$ which is a $j = 1/2$ projective representation of $G(\simeq SO(3))$. According to relation (54) and the angular momentum addition rules, $\mathcal{R}_{\text{pair}}$ is either a $j = 0$ or $j = 1$ linear irrep of $G(\simeq SO(3))$. However, because the projective representation $\mathcal{R}_\Phi(\mathcal{R}(\vec{\theta}))$ coincides with the normal-state spin rotation in (57a), the $j = 0$ irrep will preserve spin rotation and hence does not apply to the superfluid B phase. As a result, the pairing term must transform like a $j = 1$ representation under (57b). This is consistent with $\mathbf{d}(\mathbf{k}) \propto \mathbf{k}$ in this case.

### 5.2.2 Superfluid A phase of Helium-3

In the A-phase, without loss of generality, $\mathbf{d}(\mathbf{k}) = \Delta_0(k_x + ik_y)\hat{\mathbf{z}}$ [22]. The spin rotational symmetry is broken down from $S_0 = SO(3)_{\text{spin}}$ to $S = U(1)^z \rtimes \mathbb{Z}_2^x \simeq O(2)$, which is the subgroup generated by continuous spin rotations around the $\hat{\mathbf{z}}$ axis, $\mathcal{S}(\theta\hat{\mathbf{z}})$ and $\pi$ spin rotations about the $x$-axis. All possible fermion onsite symmetry groups $S_f$ are classified by $\mathcal{H}^2(S, \mathbb{Z}_2^F) = \mathcal{H}^2(O(2), \mathbb{Z}_2) = \mathbb{Z}_2^3$. Since under $\pi$ spin rotation about $x$-axis $d_z(\mathbf{k}) \to -d_z(\mathbf{k})$, the corresponding normal state transformation has to be modified by a phase rotation of $i$. No such compensating phase is thus required for spin rotation about $z$-axis. The SC state

spin rotations are implemented as

$$\text{Spin rot. } \hat{\mathcal{S}}'(\theta\hat{\mathbf{z}})\hat{c}_{\mathbf{k}s}\hat{\mathcal{S}}'(\theta\hat{\mathbf{z}})^{-1} = \left[\tilde{U}^{\mathcal{S}(\theta\hat{\mathbf{z}})}\right]^{\dagger}_{ss'}\hat{c}_{\mathbf{k}s'}, \qquad \tilde{U}^{\mathcal{S}(\theta\hat{\mathbf{z}})} = e^{i\theta\sigma_z/2}, \qquad (59a)$$

$$\text{Spin rot. } \hat{\mathcal{S}}'(\pi\hat{\mathbf{x}})\hat{c}_{\mathbf{k}s}\hat{\mathcal{S}}'(\pi\hat{\mathbf{x}})^{-1} = \left[\tilde{U}^{\mathcal{S}(\pi\hat{\mathbf{x}})}\right]^{\dagger}_{ss'}\hat{c}_{\mathbf{k}s'}, \qquad \tilde{U}^{\mathcal{S}(\pi\hat{\mathbf{x}})} = \sigma_x. \qquad (59b)$$

The central extension is characterized by $\hat{\mathcal{S}}'(\pi\hat{\mathbf{x}})^2 = 1$ and $\hat{\mathcal{S}}'(\pi\hat{\mathbf{z}})^2 = \hat{\mathcal{S}}'(\pi\hat{\mathbf{y}})^2 = (-1)^{\hat{F}}$ and correspondingly $S_f \simeq \left\{\pm\sigma_x^n e^{i\theta\sigma_z/2}|0 \le \theta < 2\pi, n = 0, 1\right\} = \left\{\sigma_x^n e^{i\xi\sigma_z}|0 \le \xi < 2\pi, n = 0, 1\right\} \simeq O(2)$.

The spatial $O(3)$ symmetry is broken down to a subgroup of $X = U(1)^z \times Z_2^I$, generated by continuous spatial rotations about $z$-axis, $\mathcal{R}(\theta\hat{\mathbf{z}})$ and inversion $\mathcal{I}$. In this case, the normal state transformations need to be modified only by compensating phase rotations. The SC state transformations are given by

$$\text{Space rot. } \hat{\mathcal{R}}'(\theta\hat{\mathbf{z}})\hat{c}_{\mathbf{k}s}\hat{\mathcal{R}}'(\theta\hat{\mathbf{z}})^{-1} = \left[\tilde{U}^{\mathcal{R}(\theta\hat{\mathbf{z}})}\right]^{\dagger}_{ss'}\hat{c}_{\mathcal{R}(\theta\hat{\mathbf{z}})\mathbf{k}s'}, \qquad \tilde{U}^{\mathcal{R}(\theta\hat{\mathbf{z}})} = e^{-i\theta/2}\cdot\mathbb{1}, \qquad (60a)$$

$$\text{Inversion } \quad \hat{\mathcal{I}}'\hat{c}_{\mathbf{k}s}\hat{\mathcal{I}}'^{-1} \quad = \left[\tilde{U}^{\mathcal{I}}\right]^{\dagger}_{ss'}\hat{c}_{-\mathbf{k}s'}, \qquad\qquad\quad \tilde{U}^{\mathcal{I}} = i\mathbb{1}. \qquad (60b)$$

In this case the fermion symmetry group $G_f \simeq (O(2) \times U(1) \times \mathbb{Z}_4)/\mathbb{Z}_2$ is a nontrivial extension of $X$ by $S_f$ satisfying $\left[\hat{\mathcal{R}}'(\pi\hat{\mathbf{z}})\right]^2 = \hat{\mathcal{I}}'^2 = (-1)^{\hat{F}}$.

# 6 Conclusion

Traditionally, the broken and unbroken symmetries of a superconductor (SC) is described by the Ginzburg-Landau theory, which characterizes the symmetry properties of all bosonic excitations therein, such as Cooper pairs. In this paper we investigate the same problem of broken and unbroken symmetries in a SC state from a viewpoint of fermionic excitations. We showed that the projective symmetry group (PSG) of fermions in a superconductor is the proper language to capture symmetry-related properties of fermionic excitations in a SC, and systematically studied the relationship between the pairing symmetry and the fermion PSGs in a superconductor. We provided a general framework in Section 5 to characterize the fermion symmetry group after the Cooper pair formation with the concept of PSG, which is a group extension of the crystalline space group $X$ by the fermion global symmetry group $S_f$ in the superconducting phase. Examples of fermion global symmetry groups include the fermion parity group $Z_2^F$ in a generic SC without spontaneous breaking of spin rotational symmetries, and $O(2)$ as in the case of superfluid A phase of Helium-3. In the case of the fermion global symmetry group $S_f$ being an Abelian group, the group extension problem can be classified by the second group cohomology, which is both conceptually clear and practically easy to compute.

When the SC and normal state share the same fermion global symmetries, i.e. in the absence of spontaneously broken spin rotational symmetries, the fermion PSG of the SC state is particularly simple: it is a central extension of the crystalline symmetry group $X$ by the fermion parity group $Z_2^F$. In this case, we can classify all fermion PSGs using elements of the 2nd cohomology group $\mathcal{H}^2(X, Z_2^F)$. Using the connection between pairing symmetry and fermion PSG discussed in section 2, we can systematically obtain all the possible pairing symmetries compatible with the PSGs as delineated in Sec. 3. A distinction was made between the case of SCs with and without spin-orbital couplings (SOCs), where in the presence of a strong SOC, crystalline symmetries of fermions in the normal state are described by a non-trivial 2-cocycle $\omega_0 \in \mathcal{H}^2(X, Z_2^F)$, and the correspondence between PSG and pairing symmetry should be shifted accordingly. Within this general framework, we calculated all the possible PSGs for all 3-dimensional point group symmetries both with and without SOCs, and establish the correspondence between PSGs and pairing symmetries of the SCs. As a demonstration of

the framework, we studied in detail the PSGs and pairing symmetries of several physically relevant systems in section 4, and hope our work would shed new lights on understandings of superconductivity in these systems. Considering the crystalline symmetry group $X$, although we have restricted our attention to point groups in this work, the case of magnetic point groups and space groups can be naturally incorporated in our general framework.

It is useful to compare the fermion PSGs in this work to PSGs initially introduced in the context of quantum spin liquids (QSLs) [21,23]. In QSLs, due to the presence of fractionalized excitations, like spinons, and emergent gauge fields, each element of the PSG is a combination of physical symmetry operation, such as a crystal symmetry $g \in X$, and local gauge rotations. In contrast, in a superconductor each element of the fermion PSG is a combination of an unbroken crystal symmetry operation $g \in X$ and a spontaneously-broken *global* symmetry operation such as a $U(1)$ charge rotation. We emphasize that our analysis does *not* involve the effects of dynamical *local* gauge fields, which have been proposed to lead to a description of superconductors as symmetry protected topological states [45] or states with $Z_2$ topological ordered states [46]. We thus treat charged superconductors and neutral paired superfluids on the same footing as systems with a broken global $U(1)$ possibly in addition to other broken symmetries.

PSGs have important implications on physical properties of a superconductor. As the PSG $G_f$ is the symmetry group of fermions in a SC, it dictates the symmetry and topological properties of all the fermionic excitations of the system and its validity extends beyond the mean-field BdG equations. Therefore, PSG can be used to classify topological superconductors in both non-interacting (i.e., admitting a mean-field description) and interacting cases. As an illustration, we discussed systems with two different kinds of symmetry groups where $G_f$ determines classifications of non-interacting topological superconductors. Moreover, as PSG establishes a link between pairing symmetry and topological properties of a system, we can utilize topological properties of the electronic excitations as a diagnosis for the pairing symmetry of a superconductor. We leave these interesting ideas for future works.

## Acknowledgments

**Funding information**  XY, SB, SL, MR and YML acknowledge support by Center for Emergent Materials at The Ohio State University, a National Science Foundation (NSF) MRSEC through NSF Award No. DMR-2011876. YML acknowledges support by grant NSF PHY-1748958 to the Kavli Institute for Theoretical Physics (KITP), and NSF PHY-2210452 to the Aspen Center for Physics.

## A   A short introduction to projective representation and 2-cocycle

In this appendix we want to elucidate the connection between the projective representation of the crystalline symmetry group as described by the mathematical object called 2-cocycle and the fermion projective symmetry group $G_f$.

The concept of PSG was first introduced in the study of quantum spin liquids [21]. In the context of quantum spin liquids, electrons can be thought of as being composed of chargons and spinons which are glued together by an $SU(2)$ gauge field. Due to the emergent gauge structures, symmetries that are represented linearly on the physical degrees of freedom are now represented only projectively on the spinons. More specifically, spin operators at site $i$ can be written as fermionic spinons: $S_i = \frac{1}{2} f_{i,\alpha}^\dagger \vec{\sigma}_{\alpha,\beta} f_{i,\beta}$. A spin Hamiltonian can be described by a mean-field theory of spinons plus gauge fluctuations. Consider the following mean-field

Hamiltonian:

$$H = \sum_{ij} [\psi_i^\dagger u_{ij} \psi_j + h.c.] + \sum_i a_0^l \psi_i^\dagger \tau^l \psi_i, \tag{A.1}$$

where $u_{ij}$'s are $2 \times 2$ matrices encoding pairing and hoppings of fermionic spinons, $\psi_i = (f_\uparrow, f_\downarrow^\dagger)^T$ are Nambu spinors.

The Hamiltonian has a local $SU(2)$ gauge redundancy: a site-dependent $SU(2)$ transformation $\psi_i \to W_i \psi_i, u_{ij} \to W_i u_{ij} W_j^\dagger$ with $W_i \in SU(2)$ which leaves both physical observables and the Hamiltonian invariant. Due to this gauge redundancy, the symmetry of the spin liquids are described by the projective symmetry group, which is defined as the collection of all combinations of symmetry elements and gauge transformations that leave the mean-field ansatz $\{u_{ij}\}$ invariant:

$$G_U U(\{u_{ij}\}) = \{u_{ij}\}, \tag{A.2}$$

$$U(\{u_{ij}\} \equiv \{\tilde{u}_{ij} = u_{U^{-1}(i), U^{-1}(j)}\}, \tag{A.3}$$

$$G_U(u_{ij}) \equiv \{\tilde{u}_{ij} = G_U(i) u_{ij} G_U^\dagger(j)\}, \tag{A.4}$$

$$G_U(i) \in SU(2), \tag{A.5}$$

where $U$ is an element of the symmetry group $SG$ of the microscopic system and $G_U$ is the $SU(2)$ gauge transformation accompanying $U$ that leaves the mean-field ansatz invariant.

To encode the emergent gauge fields at low energy for spin liquid states, we introduce the important concept of invariant gauge group (IGG) which are pure gauge group elements that leave the mean-field ansatz invariant: $W_i u_{ij} W_j^\dagger = u_{ij}$. It is clear that IGG corresponds to elements $G_U U$ in PSG where $U$ is the identity. With the concept of IGG it is now easy to describe the structure of PSG. In fact, IGG is a normal subgroup of PSG, and with the group homomorphism $\rho(G_U U) = U$ between PSG and SG, we have the following exact sequence:

$$1 \to \text{IGG} \xrightarrow{\iota} \text{PSG} \xrightarrow{\rho} \text{SG} \to 1, \tag{A.6}$$

where $\iota$ is the embedding mapping, and the exactness is ensured by the fact that $\rho(w) \equiv \mathbf{1} \in \text{SG}$ for $w \in \text{IGG}$. The structure of the PSG is now quite clear: it is the group extension of the SG by the IGG, or alternatively, SG=PSG/IGG.

Equipped with the knowledge of PSG, it is also easy to see that the problem of unbroken symmetries of the superconductor naturally fits into the general framework of PSG if we notice that the BdG Hamiltonian takes the same form as the spin liquid mean-field Hamiltonian. More precisely, as discussed in the main text, fermions in the superconductor has the symmetry group $G_f$, which is an extension of the space group $X$ by the fermion global symmetry group $S_f$ described by the short exact sequence:

$$1 \to S_f \to G_f \to X \to 1. \tag{A.7}$$

The resemblance to Eq. A.6 is immediately seen if we identify the unbroken global symmetry group $S_f$ as IGG and the fermion symmetry group $G_f$ as PSG. However, there's an important difference we need to keep in mind: in our study of superconductor, the global symmetry group $S_f$ should not be regarded as the gauge group corresponding to a fluctuating gauge field, as was in the context of spin liquids.

In general $S_f$ can be non-Abelian, and we refer to Ref. [21] for a general computation scheme to solve the extension problem by obtaining all the inequivalent projective symmetry groups $G_f$. Below let's discuss the special case of $S_f$ being Abelian, which covers most of the practical situations and is mathematically much simpler to deal with. And we will comment briefly on the case of $S_f$ being non-Abelian in the end.

In the case of $S_f$ being Abelian, the group extension of $X$ by $S_f$ can be described as an element in the second cohomology group $\mathcal{H}^2_{[\rho]}(X, S_f)$ with group actions $[\rho] : X \to \text{Aut}(S_f)$ (note that when the group action is trivial, the group extension is simply a central extension). To see this more clearly, let's label group elements in $G_f$ as $(s, g)$ with $s \in S_f, g \in X$. Now, since $S_f$ is in the center of $G_f$, we can represent the group multiplication rule in the following way:

$$(s_g, g) \times (s_h, h) = (s(g, h) s_g s_h, gh), \tag{A.8}$$

where $s(g, h)$ is a function $X \times X \to S$. The above procedure has an ambiguity since we can alternatively define $g' = \gamma_g g \in G_f$ ($\gamma_g \in S_f$) as our canonical choice of $g$. This then modifies $s(g, h)$ as:

$$s(g, h) \to s(g, h) \cdot \gamma_g \cdot \gamma_h^g \cdot \gamma_{gh}^{-1}, \tag{A.9}$$

where the superscript $g$ indicates group actions $g$ on elements in $S_f$ as described by $[\rho]$.

The $s(g, h)$'s satisfy the associativity condition if we apply three group elements in $G_f$ in two equivalent ways, which yields

$$s(g_1, g_2) s(g_1 g_2, g_3) = s(g_1, g_2 g_3) s^{g_1}(g_2, g_3). \tag{A.10}$$

The coboundary condition A.9 and the cocycle condition A.10 then define an element in $\mathcal{H}^2(X, S_f)$. Therefore we have found out that in the case of central extension, $G_f$ is uniquely determined by the 2-cocycle $s(g, h)$, which is further classified by the second cohomology group $\mathcal{H}^2(X, S_f)$.

Before proceeding, let me emphasize an important point: fermions fulfill a 1d representation of $S_f$, which we denote as $\rho_S : S_f \to U(1)$. Note that $\rho_S$ is determined by the microscopic electrons and can be viewed as a group homomorphism from $S_f$ to $\text{Image}(\rho_S)$.

Because elements in $G_f$ act on fermions in a linear way, let's consider a linear representation $\hat{U}$ of the group $G_f$. Since $S_f$ lies at the center of the group, $\hat{U}((s, 1))$ should be of the form $\rho_S(s) \times \mathbb{1}$ according to Schur's lemma and the fact that the symmetry action of $s \in S$ on fermions is given by $\rho_S$.

If we identify $U(g)$ as $\hat{U}((1, g))$, $U(g)$ would fulfill a projective representation of $X$:

$$U(g) U(h) = \hat{U}((1, g)) \hat{U}((1, h)) = \hat{U}((s(g, h), gh)) = \hat{U}((s(g, h), 1)) \hat{U}((1, gh)) = \omega(g, h) U(gh), \tag{A.11}$$

where $\omega(g, h) \equiv \rho_S(s(g, h))$ is a function $X \times X \to \text{Image}(\rho_S)$. The $\omega$ satisfies the following associativity condition if we act three consecutive symmetry operations in two equivalent ways: $g_1 g_2 g_3 = (g_1 g_2) g_3 = g_1 (g_2 g_3)$, which translates to

$$\omega(g_1, g_2) \omega(g_1 g_2, g_3) = \omega(g_1, g_2 g_3) \omega^{g_1}(g_2, g_3), \tag{A.12}$$

where the superscript $g$ on $\omega$ indicates group actions on the $U(1)$ phase induced by the group action $[\rho]$ on elements in $S_f$.

We can also multiply symmetry actions $U(g)$ by some $U(1)$ phase $\gamma_g \in \text{Image}(\rho_S)$, which then modifies $\omega$ in the following way:

$$\omega(g, h) \to \omega(g, h) \frac{\gamma_g \gamma_h^g}{\gamma_{gh}}. \tag{A.13}$$

The associativity condition (A.12) and the ambiguity (A.13) thus define a 2-cocycle in the second cohomology group $\mathcal{H}^2(X, \text{Image}(\rho_S))$. And the equation Eq.(A.11) establishes an explicit homomorphism between the projective representation of $X$ (an element in the cohomology group $\mathcal{H}^2(X, \text{Image}(\rho_S))$) and the fermion projective symmetry group $G_f$ (an element in $\mathcal{H}^2(X, S)$).

In summary, a linear representation of the fermion projective symmetry group $G_f$ can alternatively be viewed as a projective representation of the group $X$ with cocycle $\omega(g,h)$ which is an element in the cohomology group $\mathcal{H}^2(X, \text{Image}(\rho_S))$, as elucidated by Eq.(A.11).

Several remarks are in order:

1. When $S_f$ is Abelian and $\rho_S$ is injective, the two cohomology groups $\mathcal{H}^2(X, \text{Image}(\rho_S))$ and $\mathcal{H}^2(X, S_f)$ are isomorphic to each other, therefore we have sometimes used these terms interchangeably in the main text.

2. When $S_f$ is non-Abelian, $G_f$ can no longer be described by an element in the second cohomology group. If we restrict our attention to the case where the representation $\rho_S$ of $S_f$ on fermions are one dimensional, then the correspondence Eq.(A.11) still holds, enabling us to carry out calculations within this general framework.

3. When $S_f$ is non-Abelian, there are cases where the representation of $S_f$ on fermions are at least 2-dimensional, such as spin-1/2 fermions in the superfluid A phase with $S_f = O(2)$. Such cases are beyond the scope of cohomological description, and we need to solve the projective symmetry groups up to gauge equivalence on a case-by-case basis following the general procedures as described in Ref. [21].

## B How PSG constrains the pairing symmetry for all crystalline point groups

Since $G_f$ is the extension of $G$ by $Z_2^F$, we can view 1d projective representations $\mathcal{R}_\Phi(g)$ of $G$ as regular representations $\bar{\mathcal{R}}_\Phi(\hat{g}')$ for $\hat{g}' \in G_f$ with $\bar{\mathcal{R}}_\Phi(d) = -1$ ( $d \equiv (-1)^{\hat{F}}$ ) when restricted to the subgroup $X = G_f/Z_2^F$. This is confirmed by the following relation:

$$\bar{\mathcal{R}}_\Phi((\eta_g, \hat{g}'))\bar{\mathcal{R}}_\Phi((\eta_h, \hat{h}')) = \bar{\mathcal{R}}_\Phi((\eta_g \eta_h \tilde{\omega}(g,h), \hat{g}'\hat{h}')) = \omega(g,h)\bar{\mathcal{R}}_\Phi((\eta_g \eta_h, \hat{g}'\hat{h}')), \quad \text{(B.1)}$$

where we have used the fact that $Z_2^F$ is the center of $G_f$ and $\eta_g, \eta_h = \pm 1$.

Our strategy then is to first obtain the group extension $G_f \in \mathcal{H}^2(X, Z_2^F)$ and then compute the 1d irreducible representations $\bar{\mathcal{R}}_\Phi(g)$ of $G_f$ with $Z_2^F = -1$, from which we can readily obtain $\mathcal{R}_{pair}$. We used GAP computer algebra program [24] in all these calculations, which is ideally suited for the task. The results are displayed in Table 7.

## C GAP program for PSG calculation

Groups, Algorithms and Programming (GAP) is a software system designed for algebraic computations. In this section, we provide further details on how GAP is applied to the theories discussed in Sections 2, 3.

To calculate the group cohomology of point groups, we use the `HAP` and `Cryst` packages in GAP. Starting from a point group $G$, we calculate its second cohomology with coefficients in $\mathbb{Z}_2$ using the `TwoCohomologyGeneric` function, and we determine the representations of $G$ using the `Irr` function. For each cohomology class, we obtain the unique (up to coboundary equivalence) group extension $\tilde{X}_f$ of $G$ via the `FpGroupCocycle` function. From the `TwoCohomologyGeneric` function and the `FpGroupCocycle` function we can easily construct the quotient map $\tilde{X}_f \to G$. Next, we calculate the representations and characters of $\tilde{X}_f$. For each one-dimensional irrep of $\tilde{X}_f$, we simply square it and use the quotient map $\tilde{X}_f \to G$ to obtain the corresponding irrep of $G$. We then look up the group representation tables in Ref. [47] to find the corresponding $\mathcal{R}_{pair}$ as listed in the last column of Table. 7. The pseudocode is presented in Algorithm 1.

One complication in the algorithm is to obtain a complete and linearly-independent list of gauge-invariant cocycles to label the cohomology classes $\mathcal{H}^2(G, \mathbb{Z}_2)$. This is done manually by first listing all the gauge-invariant cocycles of the form $\zeta_g \equiv \omega(g, g)$ for group element $g$ satisfying $g^2 = \mathbb{1}$, and $\eta_{g,h} \equiv \frac{\omega(g,h)}{\omega(h,g)}$ for group elements $g, h$ satisfying $gh = hg$. We have tested that this list completely characterize all the cohomology classes. We then find the linearly-independent ones among these gauge-invariant cocycles to label all the cohomology classes as shown in column 3 of Table. 7.

---

**Algorithm 1** Pseudocode for calculating PSGs and corresponding pairing symmetries

---

    **Input:** Point group G
    **Output:** the (spinless) PSG $\omega_\Phi$
    **Output:** the corresponding pairing symmetry $\mathcal{R}_{pair}$
1: Coh $= \mathcal{H}^2(G, \mathbb{Z}_2)$                 //Obtain all information on the second cohomology
2: Char = Characters of irreps of $G$
3: **for** $\omega_\Phi \in$ Coh **do**
4:     $\tilde{X}_f =$ The extended group of $G$ corresponding to $\omega_\Phi$
5:     CharExt = Characters of irreps of $\tilde{X}_f$
6:     Identify the group elements of $\tilde{X}_f$ within conjugacy classes of $G$
7:     **for** $\mathcal{R}_\Phi \in$ CharExt **do**
8:         **if** $\mathcal{R}_\Phi$ is a 1D irrep **then**
9:             $\mathcal{R}_{pair} = \mathcal{R}_\Phi^2$
10:             Identify $\mathcal{R}_{pair} \in$ Char
11:         **end if**
12:     **end for**
13: **end for**
14: **Return** All matching pairs ($\omega_\Phi \in$ Coh, $\mathcal{R}_{pair} \in$ Char)

---

Table 7: Correspondence between the fermion PSG and the representation of the pairing order parameter for all the crystalline point groups. We list gauge-invariant cocycles to label different projective symmetry groups $G_f$ for superconductors both without and with spin-orbital couplings. We follow the convention in Ref. [47] to label irreducible representations $\mathcal{R}_{pair}(g)$ of the pairing order parameter. Some $G_f$ does not admit a 1d projective representation and hence the corresponding $\mathcal{R}_{pair}$ is marked as N/A. For gauge invariant cocycles, we use the following short-hand notations: $\zeta_g \equiv \omega(g,g)$, and $\eta_{g,h} \equiv \frac{\omega(g,h)}{\omega(h,g)}$.

| $X$ | $\mathcal{H}^2(X, Z_2^F)$ | Gauge-invariant 2-cocycles | No SOC (spinless) | w/ SOC (spinful) | $\mathcal{R}_{pair}(g)$ |
|---|---|---|---|---|---|
| $C_1$ | $\mathbb{Z}_1$ | $-$ | $-$ | $-$ | $A$ |
| $C_i$ | $\mathbb{Z}_2$ | $\zeta_i$ | $1$ | $1$ | $A_g$ |
| | | | $-1$ | $-1$ | $A_u$ |
| $C_2$ | $\mathbb{Z}_2$ | $\zeta_{C_2}$ | $1$ | $-1$ | $A$ |
| | | | $-1$ | $1$ | $B$ |
| $C_s$ | $\mathbb{Z}_2$ | $\zeta_{\sigma_h}$ | $1$ | $-1$ | $A'$ |
| | | | $-1$ | $1$ | $A''$ |
| $C_{2h}$ | $\mathbb{Z}_2^3$ | $(\zeta_{C_2}, \zeta_i, \zeta_{\sigma_h})$ | $(1,1,1)$ | $(-1,1,-1)$ | $A_g$ |
| | | | $(1,-1,-1)$ | $(-1,-1,1)$ | $A_u$ |
| | | | $(-1,1,-1)$ | $(1,1,1)$ | $B_g$ |
| | | | $(-1,-1,1)$ | $(1,-1,-1)$ | $B_u$ |
| | | | other cases | other cases | N/A |
| $D_2$ | $\mathbb{Z}_2^3$ | $(\zeta_{C_{2x}}, \zeta_{C_{2y}}, \zeta_{C_{2z}})$ | $(1,1,1)$ | $(-1,-1,-1)$ | $A$ |
| | | | $(-1,-1,1)$ | $(1,1,-1)$ | $B_1$ |
| | | | $(-1,1,-1)$ | $(1,-1,1)$ | $B_2$ |
| | | | $(1,-1,-1)$ | $(-1,1,1)$ | $B_3$ |
| | | | other cases | other cases | N/A |
| $C_{2v}$ | $\mathbb{Z}_2^3$ | $(\zeta_{C_2}, \zeta_{\sigma_v}, \zeta_{\sigma'_v})$ | $(1,1,1)$ | $(-1,-1,-1)$ | $A_1$ |
| | | | $(1,-1,-1)$ | $(-1,1,1)$ | $A_2$ |
| | | | $(-1,1,-1)$ | $(1,-1,1)$ | $B_1$ |
| | | | $(-1,-1,1)$ | $(1,1,-1)$ | $B_2$ |
| | | | other cases | other cases | N/A |
| $D_{2h}$ | $\mathbb{Z}_2^6$ | $(\zeta_{C_{2x}}, \zeta_{C_{2y}}, \zeta_i,$ $\eta_{C_{2x},i}, \eta_{C_{2y},i}$ $\eta_{C_{2x},C_{2y}})$ | $(1,1,1,1,1,1)$ | $(-1,-1,1,1,1,-1)$ | $A_g$ |
| | | | $(1,-1,1,1,1,1)$ | $(-1,1,1,1,1,-1)$ | $B_{3g}$ |
| | | | $(-1,1,1,1,1,1)$ | $(1,-1,1,1,1,-1)$ | $B_{2g}$ |
| | | | $(-1,-1,1,1,1,1)$ | $(1,1,1,1,1,-1)$ | $B_{1g}$ |
| | | | $(1,1,-1,1,1,1)$ | $(-1,-1,-1,1,1,-1)$ | $A_u$ |
| | | | $(1,-1,-1,1,1,1)$ | $(-1,1,-1,1,1,-1)$ | $B_{3u}$ |
| | | | $(-1,1,-1,1,1,1)$ | $(1,-1,-1,1,1,-1)$ | $B_{2u}$ |
| | | | $(-1,-1,-1,1,1,1)$ | $(1,1,-1,1,1,-1)$ | $B_{1u}$ |
| | | | other cases | other cases | N/A |
| $C_4$ | $\mathbb{Z}_2$ | $\zeta_{C_2}$ | $1$ | $-1$ | $A, B$ |
| | | | $-1$ | $1$ | $E$ |
| $S_4$ | $\mathbb{Z}_2$ | $\zeta_{C_2}$ | $1$ | $-1$ | $A, B$ |
| | | | $-1$ | $1$ | $E$ |

| $X$ | $\mathcal{H}^2(X, Z_2^F)$ | Gauge-invariant cocycles | No SOC | With SOC | $\mathcal{R}_{pair}(g)$ |
|---|---|---|---|---|---|
| $C_{4h}$ | $\mathbb{Z}_2^3$ | $(\zeta_{C_2}, \zeta_i, \eta_{C_4,i})$ | $(1,1,1)$ | $(-1,1,1)$ | $A_g, B_g$ |
| | | | $(-1,-1,1)$ | $(1,-1,1)$ | $E_u$ |
| | | | $(1,-1,1)$ | $(-1,-1,1)$ | $A_u, B_u$ |
| | | | $(-1,1,1)$ | $(1,1,1)$ | $E_g$ |
| | | | other cases | other cases | N/A |
| $D_4$ | $\mathbb{Z}_2^3$ | $(\zeta_{C_2}, \zeta_{C_2'}, \zeta_{C_2''})$ | $(1,1,1)$ | $(-1,-1,-1)$ | $A_1$ |
| | | | $(1,-1,-1)$ | $(-1,1,1)$ | $A_2$ |
| | | | $(1,1,-1)$ | $(-1,-1,1)$ | $B_1$ |
| | | | $(1,-1,1)$ | $(-1,1,-1)$ | $B_2$ |
| | | | other cases | other cases | N/A |
| $C_{4v}$ | $\mathbb{Z}_2^3$ | $(\zeta_{C_2}, \zeta_{\sigma_v}, \zeta_{\sigma_d})$ | $(1,1,1)$ | $(-1,-1,-1)$ | $A_1$ |
| | | | $(1,1,-1)$ | $(-1,-1,1)$ | $B_1$ |
| | | | $(1,-1,1)$ | $(-1,1,-1)$ | $B_2$ |
| | | | $(1,-1,-1)$ | $(-1,1,1)$ | $A_2$ |
| | | | other cases | other cases | N/A |
| $D_{2d}$ | $\mathbb{Z}_2^3$ | $(\zeta_{C_2}, \zeta_{C_2'}, \zeta_{\sigma_d})$ | $(1,1,1)$ | $(-1,-1,-1)$ | $A_1$ |
| | | | $(1,1,-1)$ | $(-1,-1,1)$ | $B_1$ |
| | | | $(1,-1,1)$ | $(-1,1,-1)$ | $B_2$ |
| | | | $(1,-1,-1)$ | $(-1,1,1)$ | $A_2$ |
| | | | other cases | other cases | N/A |
| $D_{4h}$ | $\mathbb{Z}_2^6$ | $(\zeta_{C_2'}, \zeta_{C_2''}, \zeta_i, \zeta_{C_2}, \eta_{C_2',i}, \eta_{C_2'',i})$ | $(1,1,1,1,1,1)$ | $(-1,-1,1,-1,1,1)$ | $A_{1g}$ |
| | | | $(1,1,-1,1,1,1)$ | $(-1,-1,-1,-1,1,1)$ | $A_{1u}$ |
| | | | $(1,-1,1,1,1,1)$ | $(-1,1,1,-1,1,1)$ | $B_{1g}$ |
| | | | $(1,-1,-1,1,1,1)$ | $(-1,1,-1,-1,1,1)$ | $B_{1u}$ |
| | | | $(-1,-1,1,-1,1,1)$ | $(1,1,1,1,1,1)$ | $A_{2g}$ |
| | | | $(-1,-1,-1,1,1,1)$ | $(1,1,-1,-1,1,1)$ | $A_{2u}$ |
| | | | $(-1,1,1,1,1,1)$ | $(1,-1,1,-1,1,1)$ | $B_{2g}$ |
| | | | $(-1,1,-1,1,1,1)$ | $(1,-1,-1,-1,1,1)$ | $B_{2u}$ |
| | | | other cases | other cases | N/A |
| $C_3$ | $\mathbb{Z}_1$ | $-$ | $-$ | $-$ | $A_1, E$ |
| $C_{3i}$ | $\mathbb{Z}_2$ | $\zeta_i$ | $+1$ | $+1$ | $A_g, E_g$ |
| | | | $-1$ | $-1$ | $A_u, E_u$ |
| $D_3$ | $\mathbb{Z}_2$ | $\zeta_{C_2}$ | $+1$ | $-1$ | $A_1$ |
| | | | $-1$ | $+1$ | $A_2$ |
| $C_{3v}$ | $\mathbb{Z}_2$ | $\zeta_{\sigma_v}$ | $+1$ | $-1$ | $A_1$ |
| | | | $-1$ | $+1$ | $A_2$ |
| $D_{3d}$ | $\mathbb{Z}_2^3$ | $(\zeta_{C_2'}, \zeta_i, \eta_{C_2',i})$ | $(1,1,1)$ | $(-1,1,1)$ | $A_{1g}$ |
| | | | $(1,-1,1)$ | $(-1,-1,1)$ | $A_{1u}$ |
| | | | $(-1,1,1)$ | $(1,1,1)$ | $A_{2g}$ |
| | | | $(-1,-1,1)$ | $(1,-1,1)$ | $A_{2u}$ |
| | | | other cases | other cases | N/A |
| $C_6$ | $\mathbb{Z}_2$ | $\zeta_{C_2}$ | $+1$ | $-1$ | $A, E_1$ |
| | | | $-1$ | $+1$ | $B, E_2$ |
| $C_{3h}$ | $\mathbb{Z}_2$ | $\zeta_{\sigma_h}$ | $+1$ | $-1$ | $A', E'$ |
| | | | $-1$ | $1$ | $A'', E''$ |

| $X$ | $\mathcal{H}^2(X, Z_2^F)$ | Gauge-invariant cocycles | No SOC | With SOC | $\mathcal{R}_{pair}(g)$ |
|---|---|---|---|---|---|
| $C_{6h}$ | $\mathbb{Z}_2^3$ | $(\zeta_{C_2}, \zeta_i, \eta_{C_2,i})$ | $(1,1,1)$ | $(-1,1,1)$ | $A_g, E_{1g}$ |
| | | | $(1,-1,1)$ | $(-1,-1,1)$ | $A_u, E_{1u}$ |
| | | | $(-1,1,1)$ | $(1,1,1)$ | $B_g, E_{2g}$ |
| | | | $(-1,-1,1)$ | $(1,-1,1)$ | $B_u, E_{2u}$ |
| | | | other cases | other cases | N/A |
| $D_6$ | $\mathbb{Z}_2^3$ | $(\zeta_{C_2}, \zeta_{C_2'}, \eta_{C_2,C_2'})$ | $(1,1,1)$ | $(-1,-1,-1)$ | $A_1$ |
| | | | $(1,-1,1)$ | $(-1,1,-1)$ | $A_2$ |
| | | | $(-1,1,1)$ | $(1,-1,-1)$ | $B_2$ |
| | | | $(-1,-1,1)$ | $(1,1,-1)$ | $B_1$ |
| | | | other cases | other cases | N/A |
| $C_{6v}$ | $\mathbb{Z}_2^3$ | $(\zeta_{C_2}, \zeta_{\sigma_v}, \eta_{C_2,\sigma_v})$ | $(1,1,1)$ | $(-1,-1,-1)$ | $A_1$ |
| | | | $(1,-1,1)$ | $(-1,1,-1)$ | $A_2$ |
| | | | $(-1,1,1)$ | $(1,-1,-1)$ | $B_2$ |
| | | | $(-1,-1,1)$ | $(1,1,-1)$ | $B_1$ |
| | | | other cases | other cases | N/A |
| $D_{3h}$ | $\mathbb{Z}_2^3$ | $(\zeta_{\sigma_v}, \zeta_{\sigma_h}, \eta_{\sigma_h,\sigma_v})$ | $(1,1,1)$ | $(-1,-1,-1)$ | $A_1'$ |
| | | | $(1,-1,1)$ | $(-1,1,-1)$ | $A_2''$ |
| | | | $(-1,1,1)$ | $(1,-1,-1)$ | $A_2'$ |
| | | | $(-1,-1,1)$ | $(1,1,-1)$ | $A_1''$ |
| | | | other cases | other cases | N/A |
| $D_{6h}$ | $\mathbb{Z}_2^6$ | $(\zeta_{C_2}, \zeta_{C_2'}, \zeta_i,$ $\eta_{C_2,C_2'}, \eta_{C_2,i},$ $\eta_{C_2',i})$ | $(1,1,1,1,1,1)$ | $(-1,-1,1,-1,1,1)$ | $A_{1g}$ |
| | | | $(1,1,-1,1,1,1)$ | $(-1,-1,-1,-1,1,1)$ | $A_{1u}$ |
| | | | $(1,-1,1,1,1,1)$ | $(-1,1,1,-1,1,1)$ | $A_{2g}$ |
| | | | $(1,-1,-1,1,1,1)$ | $(-1,1,-1,-1,1,1)$ | $A_{2u}$ |
| | | | $(-1,1,1,1,1,1)$ | $(1,-1,1,-1,1,1)$ | $B_{2g}$ |
| | | | $(-1,1,-1,1,1,1)$ | $(1,-1,-1,-1,1,1)$ | $B_{2u}$ |
| | | | $(-1,-1,1,1,1,1)$ | $(1,1,1,-1,1,1)$ | $B_{1g}$ |
| | | | $(-1,-1,-1,1,1,1)$ | $(1,1,-1,-1,1,1)$ | $B_{1u}$ |
| | | | other cases | other cases | N/A |
| $T$ | $\mathbb{Z}_2$ | $\zeta_{C_2}$ | $1$ | $-1$ | $A, E$ |
| | | | $-1$ | $1$ | N/A |
| $T_h$ | $\mathbb{Z}_2^2$ | $(\zeta_{C_2}, \zeta_i)$ | $(1,1)$ | $(-1,1)$ | $A_g, E_g$ |
| | | | $(1,-1)$ | $(-1,-1)$ | $A_u, E_u$ |
| | | | other cases | other cases | N/A |
| $O$ | $\mathbb{Z}_2^2$ | $(\zeta_{C_2}, \zeta_{C_2'})$ | $(1,1)$ | $(-1,-1)$ | $A_1$ |
| | | | $(1,-1)$ | $(-1,1)$ | $A_2$ |
| | | | other cases | other cases | N/A |
| $T_d$ | $\mathbb{Z}_2^2$ | $(\zeta_{C_2}, \zeta_{\sigma_d})$ | $(1,1)$ | $(-1,-1)$ | $A_1$ |
| | | | $(1,-1)$ | $(-1,1)$ | $A_2$ |
| | | | other cases | other cases | N/A |
| $O_h$ | $\mathbb{Z}_2^4$ | $(\zeta_{C_2}, \zeta_{C_2'}, \zeta_i, \eta_{i,C_2'})$ | $(1,1,1,1)$ | $(-1,-1,1,1)$ | $A_{1g}$ |
| | | | $(1,1,-1,1)$ | $(-1,-1,-1,1)$ | $A_{1u}$ |
| | | | $(1,-1,1,1)$ | $(-1,1,1,1)$ | $A_{2g}$ |
| | | | $(1,-1,-1,1)$ | $(-1,1,-1,1)$ | $A_{2u}$ |
| | | | other cases | other cases | N/A |

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
