# Peer review of "Pairing Symmetry and Fermion Projective Symmetry Groups"

_SciPost Physics, doi:SciPost Phys. 17, 161 (2024)_

## Round 1 · Referee Report · Zheng-Xin Liu (Referee 1) · 2024-7-27

Strengths

(1) the authors studied the relation between the pairing symmetries of superconductors(SCs)/superfluids(SFs) and the projective symmetry groups(PSGs). It was shown that when the paring terms carry nontrivial one-dimensional representation of the symmetry group, the symmetry class of the total BdG Hamiltonian will be changed.
(2) The PSGs for all point groups with all possible paring symmetries are provided, for cases with both strong spin-orbit coupling (SOC) or weak SOC.
(3) Since different pairing symmetries yield different PSGs, it was shown that the SCs/SFs with different pairing symmetries have different topological classifications.

Weaknesses

There are some typos. For instance, the word 'negligible' was written as 'neglible'.

Report

In this manuscript, the authors studied the relation between the pairing symmetries of superconductors(SCs)/superfluids(SFs) and the projective symmetry groups(PSGs). It was shown that when the paring terms carry nontrivial one-dimensional representation of the symmetry group, the symmetry class of the total BdG Hamiltonian will be changed. The PSGs for all point groups with all possible paring symmetries are provided, for cases with both strong spin-orbit coupling (SOC) or weak SOC. Since different pairing symmetries yield different PSGs, it was shown that the SCs/SFs with different pairing symmetries have different topological classifications. These interesting conclusions in the present study enrich the understanding of the symmetries of SCs/SFs and warrant publication. Therefore I recommend the acceptance after the following questions being addressed:

(1) The authors discussed the extension of all of the 32 crystalline point groups, but ignored the lattice translation symmetries. If one considers translational symmetry, are the conclusions in the present work still valid? It would be great if the authors can discuss this issue.

(2) The authors mentioned in equation (54) the case where the pairing terms carry multi-dimensional linear representation of the symmetry group G. This case is very interesting but was not discussed in detail. Could the authors clarify the constraints of the SCs/SFs under which the multi-dimensional representation can occur? Namely, what are the conditions that there exist a PSG (which keeps the BdG Hamiltonian invariant) compatible with the pairing terms?

(3) The pairing symmetry of SCs can be detected experimentally using SQUID. A natural question is how to detect the PSG of the corresponding SCs/SFs?

(4) In sec 4.4, the authors bridged the relation between pairing symmetry and topological classification using K theory. Is it assumed that the SCs/SFs with the PSG symmetries are fully gapped? Since the p-wave and d-wave SCs/SFs may contain nodes in the energy spectrum, will this affect the classification?

Requested changes

(1) The authors discussed the extension of all of the 32 crystalline point groups, but ignored the lattice translation symmetries. If one considers translational symmetry, are the conclusions in the present work still valid? It would be great if the authors can discuss this issue.

(2) The authors mentioned in equation (54) the case where the pairing terms carry multi-dimensional linear representation of the symmetry group G. This case is very interesting but was not discussed in detail. Could the authors clarify the constraints of the SCs/SFs under which the multi-dimensional representation can occur? Namely, what are the conditions that there exist a PSG (which keeps the BdG Hamiltonian invariant) compatible with the pairing terms?

(3) The pairing symmetry of SCs can be detected experimentally using SQUID. A natural question is how to detect the PSG of the corresponding SCs/SFs?

(4) In sec 4.4, the authors bridged the relation between pairing symmetry and topological classification using K theory. Is it assumed that the SCs/SFs with the PSG symmetries are fully gapped? Since the p-wave and d-wave SCs/SFs may contain nodes in the energy spectrum, will this affect the classification?

Recommendation

Ask for minor revision

  • validity: top
  • significance: high
  • originality: high
  • clarity: high
  • formatting: excellent
  • grammar: excellent

Author:  Xu Yang  on 2024-10-26  [id 4903]

(in reply to Report 1 by Zheng-Xin Liu on 2024-07-27)
Category:
answer to question

We thank the referee for careful review of our manuscript. We have made the requested changes and address the Referee and Editorial comments.

1. The referee writes:

The authors discussed the extension of all of the 32 crystalline point groups but ignored the lattice translation symmetries. If one considers translational symmetry, are the conclusions in the present work still valid? It would be great if the authors can discuss this issue

Our response:

Our analysis focuses on SC order parameters that are spatially uniform, where the Cooper pairs condense in a state with zero center of mass momentum. Lattice translations then act trivially on the SC state and leave the BdG Hamiltonian invariant, and it is sufficient for us to focus on point group symmetry alone. Most experimentally relevant systems exhibit spatially uniform pairing (in the absence of strong disorder). It is only in exceptional circumstances – under very limited range of external parameters in a few systems – that that one expects the SC order parameter to spontaneously break translational symmetry, e.g., in FFLO or pair density wave state. In such cases, we would need to investigate space group symmetries which we leave for future investigation.

-Changes: We now say this explicitly in the revised manuscript.

2. The referee writes:

The authors mentioned in equation (54) the case where the pairing terms carry multi-dimensional linear representation of the symmetry group G. This case is very interesting but was not discussed in detail. Could the authors clarify the constraints of the SCs/SFs under which the multi-dimensional representation can occur? Namely, what are the conditions that there exist a PSG (which keeps the BdG Hamiltonian invariant) compatible with the pairing terms?

Our response:

We thank the referee for raising this question. In order for $R_{pair}$ to be a multi-dimensional irrep. in Eq. (54), the tensor product $R_{\Phi}\otimes R_{\Phi}$ of two projective representations must be a multi-dimensional irrep. of group G. Therefore, a necessary condition for the pairing order parameter to form a multi-dimensional irrep. of symmetry group G (i.e., for $R_{pair}$ to be multi-dimensional) is that $R_{pair}$ is a multi-dimensional projective representation of group G.

-Changes: We now add a few sentences at the end of the paragraph after Eq. (54) to clarify this point.

3. The referee writes:

The pairing symmetry of SCs can be detected experimentally using SQUID. A natural question is how to detect the PSG of the corresponding SCs/SFs?

Our response:

Experimental detection of the fermion PSG is in general a difficult problem. In fact, even determining the pairing symmetry can be a nontrivial problem. For example, the problem of Sr2RuO4 is not settled [arXiv:2402.12117] even after 30 years!

Some papers that address the question of the relation between fermion PSG and experimental signatures are the following: (i) Spin-sensitive measurements on the surface as well as that for manipulation of Majorana modes by electromagnetic fields [S. Kobayashi et al., Phys. Rev. B 103 224504 (2021)]. (ii) Some of us have recently discussed the connection between PSGs and optical and Raman selection rules [S. Lu et al., Phys. Rev. B 109 245119(2024)]. (iii) In an ongoing collaboration with Seishiro Ono, we are analyzing the connection between PSGs and localized Majorana zero modes in a SC vortex.

-Changes: We now cite these three papers in our manuscript.

4. The referee writes:

In sec 4.4, the authors bridged the relation between pairing symmetry and topological classification using K theory. Is it assumed that the SCs/SFs with the PSG symmetries are fully gapped? Since the p-wave and d-wave SCs/SFs may contain nodes in the energy spectrum, will this affect the classification?

Our response:

It is indeed correct that the K theory classification applies to gapped topological SCs (or insulator). Thus, weak pairing unconventional SC with gap nodes are not part of the classification. However, there are fully gapped unconventional SCs, like the (p+ip) state in 2D and the B-phase of He3 , which are topologically non-trivial. Our analysis focus on understanding how pairing symmetry through the PSG affects the classification of such states.

-Changes: We now say this explicitly in the revised manuscript.

---

## Round 1 · Referee Report · Anonymous (Referee 2) · 2024-10-7

Strengths

(1) The authors classified possible superconducting states depending on their normal state’s crystalline symmetry by using projective symmetry group (PSG). The authors derived some general constraints on pairing symmetry, which are originated from the 2-cocycle relation between the normal state PSG and superconducting state PSG.
(2) The authors presented exhautive pairing symmetry group under all the possible crystalline symmetry of the normal state using PSG.
(3) The authors have shown that the some connection to the example of A, B phase of Helium-3 fluid.

Weaknesses

(1) The paper might engage a wider audience within the superconductivity community by emphasizing more experimentally observable insights alongside the mathematical formalism.

Report

The analysis of the PSG in this paper provides insights into the relationship between pairing symmetry and topological properties, particularly in the context of fermionic symmetries in topological superconductors (TSC). Given this, I believe that it would be beneficial to clarify whether PSG analysis can also be applied to Hamiltonians with topological defects that predict the presence of Majorana modes as the authors mentioned in the introduction section.

Requested changes

(1) How might the PSG relations be experimentally beneficial in regarding discovery of topological superconductivity? Is there any no-go theorem like statement for the discovery of TSC using PSG?

(2) While the use of the GAP program for PSG calculations is mentioned, more detailed guidance on the instruction would be helpful. Specifically, providing step-by-step instructions and key parameters would enable other researchers to more easily reproduce these results.

(3)There are some typos. For instance, the word ‘negligible’ was written as ‘neglible’.

Recommendation

Publish (meets expectations and criteria for this Journal)

  • validity: top
  • significance: high
  • originality: high
  • clarity: high
  • formatting: excellent
  • grammar: excellent

Author:  Xu Yang  on 2024-10-26  [id 4902]

(in reply to Report 2 on 2024-10-07)
Category:
answer to question

We thank the referee for the careful review of our manuscript. We have made the requested changes and address the Referee and Editorial comments.

1. The referee writes:

How might the PSG relations be experimentally beneficial in regarding discovery of topological superconductivity? Is there any no-go theorem like statement for the discovery of TSC using PSG?

Our response:

Among other things, fermion PSG gives the bulk topological classification as discussed in section 4.4 of our paper. For example, Tables 5 and 6 display the connection between pairing symmetry, fermion PSG, and topological classification of 2D and 3D topological SC for certain symmetry groups.

Such classifications can lead to no-go theorems. For instance, ongoing work on the classification of Majorana zero modes (MZM) in vortex bound states shows that a class D SC, with a two-fold rotation axis can only have protected zero modes if the order parameter is odd parity and cannot have any protected mode if the order parameter is even parity. In this forthcoming paper, we will provide an exhaustive classification of the connection between pairing symmetry, fermion PSG, and the presence of protected zero modes.

2. The referee writes:

While the use of the GAP program for PSG calculations is mentioned, more detailed guidance on the instruction would be helpful. Specifically, providing step-by-step instructions and key parameters would enable other researchers to more easily reproduce these results.

Our response:

-Changes: We now provide in Appendix C more details on how to use the GAP program for PSG calculations.

3. The referee writes:

There are some typos. For instance, the word ‘negligible’ was written as ‘neglible’.

Our response:

-Changes: We have corrected this typo and others that we found on a careful reading of our manuscript.

---

## Round 2 · Author Response

We thank both the referees for their careful review of our manuscript. We have made the requested changes and addressed the comments in the two referees' reports, respectively.

---

## Round 2 · List of Changes

1. We have added a paragraph at the end of Sec. 3 to address the case of spatial symmetry.
  2. We have added a paragraph in Section 4.4 to clarify the subject under study.
  3. We have added a paragraph about the case of multi-dimensional irreps in Sec. 5.1.
  4. We have added references 18-20 concerning the experimental detection of the fermion PSG.
  5. We have added Appendix C with details on the use of GAP program.
  6. We have corrected typos we have found upon a careful reading of the manuscript.

---

## Editorial Decision

published